# Using machine learning algorithms to identify predictors of social vulnerability in the event of a hazard: İstanbul case study

Oya Kalaycıoğlu[1,2], Serhat Emre Akhanlı[3], Emin Yahya Menteşe[4], Mehmet Kalaycıoğlu[5], and Sibel Kalaycıoğlu[6]

[1]Department of Statistical Science, University College London, London, WC1E 6BT, United Kingdom
[2]Department of Biostatistics and Medical Informatics, Bolu Abant İzzet Baysal University, Bolu, 14030, Türkiye
[3]Department of Statistics, Muğla Sıtkı Koçman University, Muğla, 48000, Türkiye
[4]Kandilli Observatory and Earthquake Research Institute, Boğaziçi University, İstanbul, 34684, Türkiye
[5]Tomorrow's Cities Research Group, City Planner, Middle East Technical University, Ankara, 06800, Türkiye
[6]Department of Sociology, Middle East Technical University, Ankara, 06800, Türkiye

**Correspondence:** Oya Kalaycioglu (oyakalaycioglu@ibu.edu.tr)

**Abstract.**

To what extent an individual or group will be affected from the damage of a hazard depends not just on their exposure to the event, but on their social vulnerability – that is, how well they are able to anticipate, cope with, resist, and recover from the impact of a hazard. Therefore, for mitigating disaster risk effectively and building a disaster-resilient society to natural hazards, it is essential that policy-makers develop an understanding of social vulnerability. This study aims to propose an optimal predictive model that allows decision-makers to identify households with high social vulnerability by using a number of easily accessible household variables. In order to develop such a model, we rely on a large data set comprising a household survey ($n = 41,093$) that was conducted to generate a social vulnerability index (SoVI) in İstanbul, Türkiye. In this study, we assessed the predictive ability of socio-economic, socio-demographic, and housing conditions on the household level social vulnerability through machine learning models. We used classification and regression tree (CART), random forest (RF), support vector machine (SVM), naive Bayes (NB), artificial neural network (ANN), k-nearest neighbours (KNN), and logistic regression to classify households with respect to their social vulnerability level, which was used as the outcome of these models. Due to the disparity of class size outcome variables, subsampling strategies were applied for dealing with imbalanced data. Among these models, ANN was found to have the optimal predictive performance for discriminating households with low and high social vulnerability when random majority under sampling was applied (Area Under the Curve (AUC): $0.813$). The results from the ANN method indicated that lack of social security, living in a squatter house and job insecurity were among the most important predictors of social vulnerability to hazards. Additionally, the level of education, the ratio of elderly persons in the household, owning a property, household size, ratio of income earners, and savings of the household were found to be associated with social vulnerability. An open access R Shiny web application was developed to visually display the performance of ML methods, important variables for the classification of households with high and low social vulnerability and the spatial distribution of the variables across İstanbul neighbourhoods. The machine learning methodology and the findings

that we present in this paper can guide decision-makers in identifying social vulnerability effectively and hence let them prioritise actions towards vulnerable groups in terms of needs prior to an event of a hazard.

## 1    Introduction

The impacts of hazards are increasing at an unprecedented rate as the exposure of communities and individuals increases and climate change amplifies the intensity of the hazards (UNDRR, 2022). Moreover, urban expansion and population growth are expected to be mostly in low and middle-income countries (Mesta et al., 2022; Schipper et al., 2016) where vulnerability to hazards is significantly high due to a lack of proper urbanization practices (e.g., construction codes, infrastructure quality and availability) and socioeconomic characteristics (e.g. poverty, lack of access to livelihoods, low level of education attainment) (Dodman et al., 2013).

In this research, we focus on the socioeconomic aspect of the vulnerability phenomenon, which will be named "social vulnerability" hereafter. Based on the vulnerability definition: "The conditions determined by physical, social, economic and environmental factors or processes which increase the susceptibility of an individual, a community, assets or systems to the impacts of hazards" by UNDRR (2022); we look at specific social factors that may increase the level of adverse impacts due to a hazard. Social vulnerability increases the risks of different social groups in relation to a set of socioeconomic conditions and needs to be determined before a particular hazard hits society (Cannon, 2008). Therefore, identification of the factors that contribute to social vulnerability is crucial for building a more resilient society (Aksha et al., 2019). In doing so, some characteristics of various layers of society come to the fore in explaining the concept of social vulnerability.

There is a critical need to assess vulnerabilities for improved preparedness and ability to recover from hazards at different scales, however, only a few studies assessed vulnerability at the individual household level in developing countries (Debesai, 2020). Within this frame, we aim to understand the factors that influence social vulnerability by utilising machine learning (ML) techniques which give us the chance to deal with big household databases. By that our target is to provide an efficient approach that can be adopted within different spatial contexts for comprehending the determinants of social vulnerability based on easily accessible databases. ML techniques are capable of handling interactions between variables, thus the proposed approach considers interactions between factors to reflect the multidimensional and complex nature of social vulnerability. We demonstrate this approach to the İstanbul case study area in which we benefit from a previous social vulnerability study to test our methodology at household level. For building ML models, we rely on a large data set of a previous study comprising a household survey ($n = 41,093$) and pre-constructed social vulnerability index (SoVI) of these households. We consider the SoVI scores as an indication of the social vulnerability level for each household, and our focus in this study is to assess to what extent the pre-constructed SoVI (and hence the social vulnerability of the households) can be predicted with machine learning techniques using household data that are available within databases of various institutions and public authorities.

This study contributes to disaster risk research in several aspects. First, we propose a methodology to identify the descriptors of social vulnerability, which is generic enough to be adopted for any spatial context. The proposed method extracts representative predictors for social vulnerability which are accessible in most spatial contexts around the world. Second, we introduce

ML algorithms into vulnerability assessment practices which is a relatively overlooked aspect as a method in the disaster risk discipline. It is seen that ML algorithms can be used efficiently to overcome the complexity of the social vulnerability concept, particularly with large data sets. Thirdly, since there are only a limited number of studies which assesses vulnerability at the household level (particularly in developing countries) (Debesai, 2020), our method is an attempt to contribute to the literature by bringing in a more precise approach to estimating social vulnerability in a household scale.

This paper is structured into four following sections: i) context and motivation for this study, which involves a literature review on the social vulnerability context and the approaches developed to measure it, followed by our motivation on why we chose machine learning techniques as an approach to identify the descriptors of social vulnerability (Sect. 2) 2) ii) the materials and methods applied within our research (Sect. 3) iii) the results that came out as a consequence of our methodology applied (Sect. 4) and iv) conclusions and discussions where we present our findings based on the results and discuss the limitations and

rooms for improvement in our approach (Sect. 5, 6, 7).

## 2   Background for Social Vulnerability Assessment

The social, political and economic characteristics of individuals influence their status of being exposed to disasters (Cutter et al., 2009). Therefore, the human dimension has become an increasingly popular topic in disaster risk research for comprehensively assessing and understanding the potential impacts of natural hazards (Shen et al., 2018). In this regard, social science research in

the hazard domain is shaped around questions such as "Which factors influence the adoption of individuals to hazards?", "Why do people prefer to live in hazardous areas?", and "How the individuals' risk perception influences their behaviour?" (Burton et al., 2018). Answers to these questions could help to understand social indicators of vulnerability, and they explain why people with similar levels of exposure may experience very different levels of adverse impact. Social indicators of vulnerability were studied extensively in the literature (e.g., Aksha et al. (2019); Fatemi et al. (2017); Cannon (2008); Cutter et al. (2003);

Wang and Sebastian (2021)). Within these studies, social vulnerability expands over a diverse range of social, individual, and sometimes spatial characteristics.

Just to mention a few, disability, for example, is one of the most common indicators within social vulnerability literature, in which it is emphasised that disabled people are more disadvantaged in terms of coping against the implications of hazards compared to non-disabled individuals. It is also empirically known that the death rate of disabled people is higher in large-scale

disasters such as earthquakes, floods, and tsunamis (Stough and Kelman, 2018; Peek and Stough, 2010). Within demographical components, gender is also one of the most commonly used ones as women are considered more vulnerable to hazards compared to men (Llorente-Marrón et al., 2020; Martins et al., 2012; Fekete, 2009). With respect to the age dimension, it is acknowledged that children and especially elderly people over 65 who live alone are age groups that can be more affected by any disaster (e.g., Fatemi et al. (2017)). The responses of children, the elderly, the disabled, and patients to a hazard may not

be the same as those of young, healthy people (Chou et al., 2004).

Besides demographic properties, the characteristics that determine the socioeconomic level such as income, employment status, social security, and household size, have an influence on the level of vulnerability (e.g., Chen et al. (2013); Holand

et al. (2011); Evans and Kantrowitz (2002)). Enarson et al. (2018) showed that the distribution of labour affects the impact of disasters on mortality and morbidity. It must also be noted that socioeconomic status is mostly accompanied by "education level" which denotes the highest education degree a person has. In several studies, it is implied that higher education level leads to more ability to cope and/or resist hazards, as higher education level enables higher-income jobs and wealthier life (e.g., Wisner and Luce (1993); Armaş (2008)).

In addition to socioeconomic and demographic properties, in some studies, the physical environment is also considered an indicator of social vulnerability, where the infrastructure quality, availability and access to public resources such as transportation, education and health facilities are incorporated within the concept (e.g., de Oliveira Mendes (2009); Cutter et al. (2000); Holand and Lujala (2013)). It is assumed that the lack of those opportunities increases the social vulnerability of the individuals within the area of interest.

In this context, it is seen that descriptors for social vulnerability to hazards are mainly grouped under 3 dimensions: i) demographics, ii) socioeconomics, and iii) the physical environment. More detailed reviews on social vulnerability indicators can be found at (Nor Diana et al., 2021; Fekete, 2009; Fatemi et al., 2017).

Although there is more or less a consensus on the indicators of social vulnerability, measuring it is challenging due to the complexity of the concept and its latent nature (Birkmann and Wisner, 2006). To quantify social vulnerability as a single metric value, three main statistical modelling approaches are employed: inductive, deductive and hierarchical. Inductive models combine a set of large indicators into latent factors, and then sum these factors to construct a single index score for social vulnerability. Deductive models contain fewer indicators which are normalised and summed to construct the index score. Hierarchical designs aggregate indicators into groups (sub-indices) that share an underlying dimension of vulnerability. These sub-indices are then aggregated to construct a vulnerability index. The methodological comparison of these designs and various approaches to constructing a social vulnerability index are reviewed by various authors, by various authors (e.g., Tate (2012); Rufat et al. (2019); Bakkensen et al. (2017)).

Among these approaches, the social vulnerability index (SoVI) developed by Cutter et al. (2003) has been one of the most commonly used tools to quantify vulnerability (6840 citations according to Google Scholar by 1st April 2023). In the aforementioned study, SoVI was constructed by factor analysis based on principal components analysis (PCA) in the U.S. County scale based on 42 vulnerability variables. In Cutter et al. (2003), where the data from areal divisions (U.S. Counties) are used, a total of 11 factors were obtained which explains $76.4\%$ of the variance in social vulnerability in the U.S. counties. The SoVI scores were calculated by summing the raw metrics for each county, where the higher and lower scores represent high and low social vulnerability, respectively. Various studies thereafter assessed the indicators that could be used to measure social vulnerability for a certain location and time frame (Holand et al., 2011; Bergstrand et al., 2015; Fatemi et al., 2017; Rufat et al., 2019; Spielman et al., 2020; Mahbubur Rahman et al., 2022). It can be suggested that there is almost a consensus between those studies where social vulnerability is defined as a function of gender, health status and access to healthcare, poverty, age, property ownership, and socio-economic indicators (Kalaycioglu et al., 2006). For the SoVI which was constructed in İstanbul in 2018 similar variables and categories were used with reference to Cutter et al. (2003) but the data was collected via a household survey (for more information on variables see Sect. 3 and Supplementary File 1).

The inductive factor analytic framework proposed by Cutter et al. (2003) to measure social vulnerability has been widely adopted in many studies (e.g., Aksha et al. (2019); Chen et al. (2013); Rabby et al. (2019); Guillard-Gonçalves et al. (2015); Krishnan et al. (2019); Roncancio et al. (2020); Wang et al. (2022)). SoVI is a valuable tool not only for academics but also for policy-makers and governmental bodies as it allows making spatial assessments, that enables comparison of different spatial entities such as counties, districts, and neighbourhoods with respect to their social vulnerability level (e.g., Spielman et al. (2020); U.S. Environmental Protection Agency (2015); Emrich et al. (2014); Dunning and Durden (2011); Flanagan et al. (2011)). Although SoVI is used in many studies, the vulnerability research which assesses household-level social vulnerability is limited (Liu and Li, 2016; Wilson, 2019; Tasnuva et al., 2021).

Despite the common usage of SoVI and its advantages, various studies have shown that the prediction of social vulnerability can be enhanced by empirical modelling utilising historical event data and intensity measures for the given hazard (Wang and Sebastian, 2021; Wang et al., 2021; Bjarnadottir et al., 2011). Relying on empirical data can be considered a more realistic approach for estimating the social vulnerability of a given entity (compared to SoVI); however, the high dependence on data may become an obstacle, particularly for contexts where data scarcity is in place or data sharing protocols are missing. Another drawback of such an approach is that; when catastrophic hazard occurrence is rare, the policy-makers can underestimate the impacts of a major hazard event, if they rely on historical data from the smaller-scale hazardous events where the losses are much less due to infrastructural investments. Thus, data scarcity and rare occurrence of major hazards make it challenging to use historic data for a hazard-driven social vulnerability research.

In this respect, SoVI scores are commonly used as a proxy of social vulnerability, which is independent of empirical data, which enables to develop a more generic methodology that can be applied in different contexts. Within this scope, there are numerous studies that have examined the factors relating to social vulnerability in a hazard, by using either descriptive statistics (Yücel and Arun, 2010; Walker et al., 2019), or traditional data analysis tools, such as linear or logistic regression (Fekete, 2009; Noriega and Ludwig, 2012; Syed and Kumar Routray, 2014; Llorente-Marrón et al., 2020; Mtintsilana et al., 2022). While the former lacks the incorporation of the relationships between the vulnerability indicators, the latter relies heavily on data assumptions. In contrast, machine learning algorithms allow for a larger number of predictors, can handle complex interactions between predictors, can model nonlinear relationships and do not make any distributional assumptions regarding the data (Ryo and Rillig, 2017). In quantitative social research, particularly with large-scale survey data where relationships between socio-demographic and socio-economic variables cannot be ignored, there is an emerging interest in using ML methods for making predictions (Buskirk et al., 2018).

A relatively small number of researchers have opted to use ML methodology over traditional statistical techniques in vulnerability research (Table 1), and indeed a detailed model-based assessment of the predictors of social vulnerability to hazards seem lacking. The few studies that employ ML techniques were based on larger sampling units such as districts, neighbourhoods, or communities, in contrast to our study which was based on a household scale. Due to the low number of studies and significant variation in their methodology, scale level and outcome type, it is difficult to make model-based recommendations. Moreover, the performances of various ML methods are rarely compared in terms of their predictive accuracy for social vulnerability in hazards (Yoon and Jeong, 2016).

**Table 1.** Studies that assess factors related to social vulnerability using ML models.

| Study | Type of Hazard | Region | Scale Level | ML Model | Outcome | Predictors |
|---|---|---|---|---|---|---|
| (Alizadeh et al., 2018) | Earthquake | Tabriz, Iran | Municipality zones | ANN | 5-category SVI | 7 regional indicators such as densities of the population, men, women, literate people, household, employed, and unemployed people |
| (Dwyer et al., 2004) | Earthquake | Perth city, Australia | Households | CART | 2-category SV class variable, assessed with a risk perception questionnaire applied to 1100 individuals | 15 indicators related to demographic and economic household attributes |
| (Yoon and Jeong, 2016) | Any single hazard | South Korea | Local communities | Random Forest, Cubist | Community vulnerability, assessed with indicators related to economic damage | 12 indicators including social, economic, and natural environment and built environment |
| (Abarca-Alvarez et al., 2019) | Any single hazard | Andalusia | Dwelling units | CART | 2-category SV class variable, which is obtained from previous database | 66 indicators of the demographic, social, labour, facilities, and services, etc., dimensions. |

SV: Social Vulnerability, CART: Classification and Regression Trees, ANN: Artificial Neural Network

## 3 Materials and Methods

In our study, we attempt to contribute to social vulnerability research by identifying the most important factors that contribute
to the prediction of social vulnerability of households by using the ML approaches. In this regard, we address the following
research questions: (1) What is the best-performing ML method for the prediction of social vulnerability? (2) What are the most
influential predictors associated with social vulnerability? We posit that, when large data sets are available at the household
level, the models developed based on ML algorithms have the potential to predict socially vulnerable households with high
accuracy.

As an indication of hazard-related social vulnerability, we have adopted SoVI which was previously constructed in İstanbul
in 2017 (IMM, 2018; Menteşe et al., 2019). In this paper we do not intend to discuss the SoVI scores or the methodology of
this previous study; but instead, we consider the SoVI scores as a proxy of the social vulnerability state for each household.

We assessed to what extent the pre-constructed SoVI (and hence the social vulnerability of the households) can be predicted with machine learning techniques using quantifiable household variables data (such as socio-economic and socio-demographic characteristics and housing conditions) that are assumed to be available within publicly accessible databases provided by statistical institutes of central government agencies or local public authorities. Thus, we aimed at presenting an approach that can reduce the time and economic burden that decision-makers can spend collecting data and modeling to identify households with high social vulnerability.

## 3.1 Study Area

Türkiye is in a region that is prone to natural hazards where a large-scale disaster happens every seven to eight years (Baris, 2009). Among the different types of disasters, earthquakes are responsible for the most extensive losses in terms of both human life and property, accounting for the $60\%$ of disaster-related fatalities in Türkiye (AFAD, 2019). Following the earthquakes, landslides (which mostly take the form of rock falls, slides or flows, or mass movements), floods, snow avalanches, and large-scale wildfires are amongst the most commonly occurring hazardous events that have adverse impacts on human lives, as well as the environment and economy (AFAD, 2019; Çolak and Sunar, 2020). Our case study area İstanbul city is also prone to hazardous events, such as earthquakes, flooding, landslides, tsunamis, and extreme weather events (Menteşe et al., 2022). However, our site selection is not only related to İstanbul's location in a hazard-prone area but mostly related to its high population density and high level of economic investments that increase the expected losses of possible hazards in the city. İstanbul is the 15. most populated city in the world, with a population of approximately 16 million, and it is also the largest metropolitan city in Türkiye (WUP, 2023). After the 1930s, the city of İstanbul grew steadily and became the heart of Türkiye's economy, producing almost $31\%$ of the national GDP in 2021 (OECD, 2021). In the last century, the economic growth triggering mass migration to the city induced uncontrolled illegal housing with low-quality building materials in hazardous areas (Taubenböck et al., 2006). Additionally, building codes were updated in 1997, and before that, even if legally constructed, buildings were built with less stringent building codes which do not consider disaster risk (Atun and Menoni, 2014). This rapid and uncontrolled urban growth increased vulnerability to hazards in the city (Green, 2008). Hence, our study area is selected as a suitable setting for our research on social vulnerability because it is a hazard-prone zone with high population density and poor-quality housing.

## 3.2 Data source: Social vulnerability research in İstanbul in 2017

### 3.2.1 Survey sampling method and application

To provide a basis for the social vulnerability analysis, a large-scale household survey was carried out by İstanbul Metropolitan Municipality (IMM) in 2017 to assess the disaster-related social vulnerability of the households in İstanbul. The variables used in this research were in line with the social science and disaster literature, where such research is focused generally on the social factors that increase or decrease the impact of specific hazard events on the local population. The authors of this study were given permission to use this survey data after the data were fully anonymised. The exact number of surveys is

41,093 households covering 955 sub-districts/neighbourhoods, with residential occupation expanding over the whole jurisdiction boundaries of the metropolitan municipality of İstanbul (IMM, 2018). The households were randomly selected from the Address Based Population Registration System Database of the Turkish Statistical Institute using the proportionate stratified sampling method. All 955 neighbourhoods within 39 districts of İstanbul were taken as strata, then households were randomly selected from each neighbourhood. The number of households in each neighbourhood taken is proportional to the neighbourhood population. The survey was conducted via face-to-face interviews with one household member, ages between 18 and 70 and capable of giving relevant and accurate information about the household. The verbal and written informed consents were obtained from the participants during the data collection stage.

### 3.2.2 Construction of SoVI

SoVI scores of the selected households were calculated using Cutter's factor analytic framework (Cutter et al., 2003) in social vulnerability research funded and being used by IMM, as explained by Menteşe et al. (2019) and Supplementary File 1. To date, this work by the IMM has been the most comprehensive study for assessing the social vulnerability of households in the event of a hazard, which was originally constructed for earthquake-induced disasters as the most probable major hazard for İstanbul. It considers the concept of social vulnerability as a state that arises from the lack of capacity of society and individuals to cope with natural hazards. The concept further includes the perception of and preparedness for risk and the measures taken against the risk, as well as cultural values and socio-economic status. To construct SoVI, 53 indicators within 7 variable clusters (socio-demography, socio-economy, access to health services, social solidarity, risk perception and actions taken to reduce risk and values) were used as they are regarded to be related to social vulnerability. The indicators and variable clusters were selected following extensive literature reviews and expert judgement with a specific focus on earthquake hazards (IMM, 2018). In the theoretical framework, social vulnerability is considered to be independent of hazard type, and exposure zones to any or all hazards are combined with SoVI to create place vulnerability (Cutter et al., 2009). Hence the earthquake-related (as the major hazard in İstanbul) data collected in this household survey and the indicators used for SoVI are also assumed to explain other hazard events as well.

Here we note that it is quite challenging to access/find quality empirical information regarding disaster-related topics in Türkiye as in many developing countries and the global south context. Information related to historical data on disaster impact/losses/recovery is mostly not in place for smaller regional units in Türkiye, then even if it is there (gathered by related institutions), it is not shared. Therefore, Cutter et al. (2003) index-based methodology to represent social vulnerability was opted for constructing SoVI in the previous study by IMM.

### 3.3 Outcome of the machine learning models: Household-level social vulnerability

In this study, we relied on the pre-constructed SoVI as an indication of the social vulnerability of the households. By that, we used SoVI as the outcome of the machine learning models we tested. SoVI score does not have any unit and rather than its absolute value, its importance lies within its comparative value across various households (Cutter and Finch, 2008). Various authors dichotomised social vulnerability index scores in their research for both ease of interpretation and to identify those

most vulnerable (Dwyer et al., 2004; Abarca-Alvarez et al., 2019; Basile Ibrahim et al., 2021; Mtintsilana et al., 2022). In this research, we also aimed to discriminate between the most vulnerable households and all others. Therefore, we defined households with high social vulnerability (SV) as those with SoVI scores $+1$ standard deviation from the mean which corresponds to $17.2\%$ of the households, whereas the rest of the households were deemed as low SV. Thus, a binary variable (with an approximate imbalance ratio of $1/5$ in favour of low SV) was generated as an indication of social vulnerability level, which in turn was used as the primary outcome for all the further analyses presented in this paper. Further, from the statistical point of view, we preferred to dichotomise the outcome rather than using it as a multi-category variable, as the available performance metrics for a multi-class confusion matrix are limited compared to a binary classification problem and the complexity of analysis increases with the increase in a number of classes (Markoulidakis et al., 2021). Therefore, in accordance with our motivation and for interpretive reasons we used SoVI as a binary outcome.

### 3.4 Predictors of the machine learning models and data pre-processing

We have restricted the variables that are used in the ML models as input variables to quantifiable predictors which can be obtained from various institutional databases without requiring a household-based survey that is costly and time intensive. These quantifiable predictors are related to the socio-demography and socio-economy of the households as well as housing information. The list of institutions to which the variables used in this study are related is given in Supplementary File 2. Here we note that, although the household data used in the IMM (2018) to construct SoVI is focused on earthquakes; the indicators used for social vulnerability classification in the present study can be implemented in a more generic way to assess the possible impact of social vulnerability to other hazards.

Prior to model development, the predictors were prepared in terms of data representation, standardisation and feature selection. As the predictors represent household characteristics, they were sought at the household level. As stated by Akhanli and Hennig (2020), data representation is about enabling better interpretation of the relevant information. Therefore, the predictors which are measured at the household level, such as the number of women, men, $< 5$ years olds, $> 65$ years old, and the number of income earners were taken in proportion to the given household's size (HhS). Then, in order to make the variation of continuous variables comparable, these variables were standardized into the same scale with unit variance standardization (Hennig and Liao, 2013). For the final step, we used feature selection prior to processing the data and we identified the predictors with near zero variance, as the predictors which take only one value may cause numerical problems during resampling (Kuhn, 2008). The set of 26 variables used for model building is presented in Table 2, along with their relevance in relation to the objectives of our study.

### 3.5 Machine learning methods

We developed models for the classification of households in terms of their social vulnerability in the event of an earthquake using six supervised machine learning algorithms: classification and regression tree (CART), random forest (RF), artificial neural network (ANN), support vector machine (SVM), naïve Bayes (NB), k-nearest neighbours (KNN). The predictive performances of these ML models are compared to that of the logistic regression (LR) model, which is a traditional statistical technique used

**Table 2.** Predictors used in ML model building for prediction household level social vulnerability.

| Themes | Variable | Definition of a variable or survey question |
|---|---|---|
| Socio-Demographic | Household size | Number of people living in the house (HhS) (Range: $1 - 14$) |
| | Average age | Average age of the household members in years (Range: $8.8 - 85$) |
| | Number of women/HhS | Ratio of women in the household (Range: $0 - 1$) |
| | Number of men/HhS | Ratio of men in the household (Range: $0 - 1$) |
| | Number of $< 5$ year olds/HhS | Ratio of $< 5$ years old children in the household (Range: $0 - 0.67$) |
| | Number of $> 65$ years of age/HhS | Ratio of over 65 years old individuals in the household (Range: $0 - 0.1$) |
| | Average education | Average years of education of the household members who are over 15 years old (Range: $0 - 17$) |
| | Social security | Are there any household members with social security? (yes/no) |
| Health | Health insurance | Are there any household members with health security or insurance? (yes/no) |
| | Disability | Are there any disabled or elderly persons who need care in the Hh? (yes/no) |
| | Health access | Do you have any hospital/health centre within close proximity to your house? (yes/no) |
| Socio-Economic | Number of income earners/HhS | Ratio of the number of income earners in the household (Range: $0 - 2$) |
| | Regular salary income | Are there any household members who have regular salary income? (yes/no) |
| | Pension income | Are there any household members who earn pension income? (yes/no) |
| | Rent income | Are there any household members who earn income from rent? (yes/no) |
| | Income support from public authorities | Are there any household members who receive income support from public authorities? (yes/no) |
| | Job Insecurity | Are there any household members who have job insecurity? i.e., unregistered informal work, unemployment (yes/no) |
| | House ownership | Do any of the household members own the house of your residence? (yes/no) |
| | Type of the house | What is the type of the home of your residence? (apartment flat, squatter house, detached house, gatekeepers lodge) |
| | Natural gas heating | Do you have natural gas heating at the home of your residence? (yes/no) |
| | Own house in İstanbul | Are there any household members who own a house in İstanbul, other than the home of residence? (yes/no) |
| | Own land in İstanbul | Are there any household members who own land in İstanbul? (yes/no) |
| | Own house out of İstanbul | Are there any household members who own house outside İstanbul? (yes/no) |
| | Own land out of İstanbul | Are there any household members who own land outside İstanbul? (yes/no) |
| | Saving | Are there any household members who have savings to use for emergency situations? (yes/no) |
| | Debt | Are there any household members who have debt to banks (inc. credits, bank loans, etc.)? (yes/no) |

for binary classification. Supervised ML adopts an algorithm to learn the mapping function from the input variables to the output variable and it is suited well to classification problems. Models were developed using the variable set in Table 2 as the input variables, while a binary indicator of the social vulnerability level of each household was the output variable. We developed a prediction model using $90\%$ of the data set to train the underlying algorithm, while $10\%$ was held back as independent testing data for evaluating the performance of the models. We note that these algorithms have different tuning parameters. For different tuning parameter alternatives, the choice of the optimal tuning parameter was determined by the largest area under the curve (AUC) value of the receiver operating characteristic (ROC) curve using the automated grid search. The details regarding the machine learning models and R software packages used for the analysis are provided in Supplementary File 3. The workflow for the model building is shown in Fig 1.

## 3.6 Data level pre-processing

### 3.6.1 Resampling techniques

Repeated cross-validation (RCV) and bootstrap resampling procedures were used to draw multiple subsamples from the original data to build machine learning models on the training data and to validate the models, in each instance, on the data that were excluded from the subsample. The tuning parameters were selected as 5-fold with 4 repetitions for repeated cross-validation and 20 repetitions for bootstrap, resulting in the same amount of resampling. The number of resampling repetitions was kept low to diminish the computational time burden.

### 3.6.2 Subsampling for the imbalanced class variables

A data set is said to be imbalanced when the classification categories are not represented equally (Lin and Nguyen, 2020). In our study, the social vulnerability data set consists of imbalanced class variables, in which the "high SV" class has a lower frequency compared to the "low SV" class. The imbalance ratio of these two classes was approximately $1/5$. The main challenge of the imbalance problem in standard machine learning algorithms is that the minority classes can be overlooked and weighed down by the majority one (Ramyachitra and Manikandan, 2014). In order to address this issue, we used various subsampling approaches during the data pre-processing steps as explained below:

(i) *Random majority under-sampling (Under)*: Under-sampling randomly samples from the majority class and returns a subsample which has the same size as the minority class, thus ensuring the majority class prevalence is equal to that of minority one for subsequent modelling (Batista et al., 2004). For instance, assume a binary class variable in which $90\%$ of training set samples belong to the majority class, while the remaining $10\%$ are in the minority class. Under-sampling will randomly subsample from the majority class such that its prevalence is $10\%$. As a result, only $20\%$ of the total training set will be used for the classification model. While balancing the class variable, however in some cases this approach may remove many important or otherwise influential data points prior to modelling.

(ii) *Over-sampling*: Three different over-sampling strategies were applied:

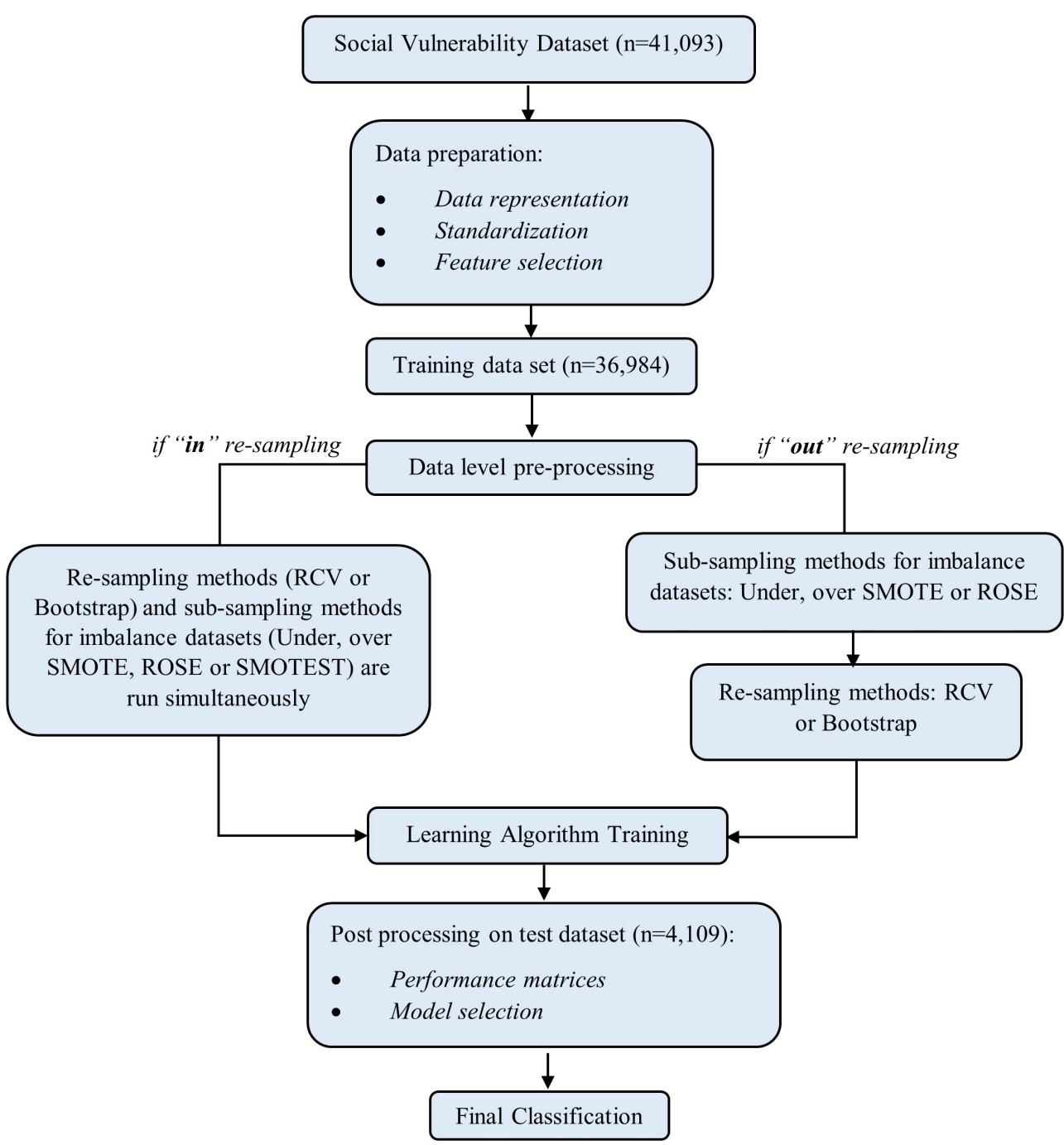

**Figure 1.** Machine learning flowchart for data processing and model development.

– *Random minority over-sampling (Over)*: It aims to balance the distribution of the class variable by taking random replicates of the minority class (Batista et al., 2004). Although it helps to improve the accuracy of classification in imbalanced data sets, it is prone to overfitting and computational problems when the data set is large (Maheshwari et al., 2017).

– *Synthetic Minority Over-sampling Technique (SMOTE)*: It creates artificial minority examples by interpolating between randomly selected examples of the minority class and their nearest neighbours (Chawla et al., 2002). It attempts to avoid the overfitting problem by using new synthetic minority class examples instead of replicating minority samples.

– *Random Over-Sampling Examples (ROSE)*: It generates artificial balanced samples according to a smoothed boot-strap approach and aids in the phases of estimation and accuracy evaluation of a classification algorithm in the presence of an imbalanced class variable (Menardi and Torelli, 2014).

The above procedures are independent of resampling methods such as repeated cross-validation and bootstrap. On the other hand, these subsampling procedures can also be performed for the resampling techniques, so that subsampling is conducted inside of resampling. In this paper, when subsampling procedures are performed outside of resampling techniques it is referred to as "out sampling", otherwise it is expressed as "in sampling".

One could also consider creating a custom-made subsampling procedure. In this respect, we also apply the transformed version of SMOTE that use 10 nearest neighbours instead of the default of 5 by adopting a simple wrapper function, which we call the "SMOTEST". Note that the SMOTEST function is only performed inside the resampling (Kuhn and Johnson, 2013).

## 3.7  Statistical analysis and model performance assessment

The characteristics of the study population were summarised using descriptive statistics. Pearson's chi-square tests were used to compare categorical variables, and independent samples t-tests or non-parametric Mann Whitney U tests were used to compare continuous variables between the high and low SV groups depending on the data distribution. In studies with large sample sizes, in addition to $p$-values, it is also relevant to provide effect sizes as it can help decide whether the difference found is meaningful or not (Bakker et al., 2019). Thus, we have reported effect sizes in the univariate comparisons that measure the strength of the relationship between two variables along with the $p$-values to assess whether the effect of a variable is real and large enough to be useful or not. Cohen's $d$ statistic with sample size adjustment was used for normally distributed continuous variables, Cohen's $r$ value which is calculated by dividing the z value obtained from the Mann Whitney test by the square root of the sample size was used for non-normally distributed variables, and Cramer's $V$ is used for categorical variables (Fritz et al., 2012).

For various machine learning applications confusion matrices were generated. Sensitivity, specificity, and accuracy with 95% confidence intervals (CIs) were calculated for LR and each ML algorithms using different resampling and subsampling techniques. The models were fitted with two different resampling strategies and eight subsampling techniques. In addition,

we fitted the models to the raw data without any subsampling, and thus we obtained results for 18 combinations of various sampling strategies for each ML algorithm.

In line with the objective of the study, we compared the methods in terms of their success in identifying the households with high social vulnerability, which is the minority class with a smaller prevalence in our study. Therefore, we used sensitivity (true positives/(true positives+false negatives)) as the primary measure for assessing the model performance. As an indication of model accuracy, we used balanced accuracy ((sensitivity + specificity)/2) which performs better on imbalanced data sets. We identified the best performing method as the one with the highest sensitivity and balanced accuracy, provided that the AUC of the ROC curve is greater than 0.7 and the model could be considered acceptable to discriminate households with high SV from those with low SV (Hosmer et al., 2013).

The sensitivity and specificity of the best-performing method with those of other methods were compared with pairwise comparisons using McNemar's chi-square test (Kim and Lee, 2017). In addition, AUC comparisons were performed using DeLong chi-square statistics (DeLong et al., 1988). Bonferroni adjustment was applied in these pairwise comparisons of ML methods and $\alpha < 0.05/7 = 0.007$ was considered as an indication of a statistically significant difference in terms of performance metrics between two methods.

## 3.8 Variable importance analysis

As the final step of our analysis, the important variables of each model were assessed. Analysing variable importance is important in machine learning applications because it assists in the interpretation of the model. It can be performed in two ways: (1) by using a model-based approach which computes the contribution of the predictor variables to the model, or (2) by evaluating the importance of predictors individually by conducting an ROC curve analysis for each predictor in turn (Kuhn, 2008). How to choose which approach to use depends on which ML model was employed.

Logistic regression models rank the variables according to standardised coefficients. The regression coefficients of continuous variables are standardised by dividing each coefficient by a value twice its standard deviation, as explained in Gelman (2008). The coefficients for factor variables are left unchanged. The relative importance of the independent variables for ANN models are computed by Garson weights (Garson, 1991), which identify all weighted connections between the nodes of interest. In this context, the weights connecting the variables can be thought of as similar to coefficients in a regression model and are used to describe the relationships between outcome and predictor variables. In random forests, variable importance analysis is based on the prediction accuracy of the model. The average differences between the out-of-bag errors before and after permuting each predictor variable over all trees are calculated as an indication of the importance of a variable. The underlying idea is that a permutation of an important variable reduces the accuracy of the model more strongly than a permutation of an unimportant variable (Couronné et al., 2018). On the other hand, another tree-based method, CART, does not use the permutation technique for measuring variable importance as it is trained on a single decision tree. Instead, CART depends on an impurity metric - which is often called the 'Gini-index' - for determining the importance of a variable when the outcome is categorical (Krzywinski and Altman, 2017).

For classification models (e.g., NB, KNN and SVM) there is no available model-specific variable importance metric. Rather, these models calculate the area under the ROC curve for each predictor variable, and this AUC statistic is considered as the measure of variable importance (Kuhn, 2008).

### 3.9 Open-access R Shiny web application

An open-access R Shiny web application was created for visualising summary statistics and predictive performances of the LR and ML methods for the classification of households in terms of their social vulnerability level. Users are able to examine the distribution of the characteristics of the households with high and low social vulnerability, compare the performances of ML and subsampling methods based on user-defined evaluation criteria, assess variable importance rankings for each ML method and obtain the area-based calculations of the variables in the İstanbul map. The R Shiny web application is freely available online and can be accessed at https://oyakalaycioglu.shinyapps.io/Social_Vulnerability/. The components of this R Shiny application are presented in detail in Fig. 2. All analyses were performed in the statistical programming environment R version 4.0.3 (Team, 2021) and the machine learning model development was carried out using the R caret package (Kuhn, 2008). The spatial distribution of the important predictors within the city scale was expressed via the 3.10 version of QGIS software (QGIS, 2021).

## 4   Results

### 4.1   Descriptive statistics

The prevalence of households with high social vulnerability to a possible hazard in İstanbul was $7,052$ ($17.2\%$) among $41,093$ households. The median household size was $3$, with values ranging from $1$ to $14$ residents, and the median average age of the households varied between $8.8$ to $85$ years with the median being $35.5$. The median of the average education was $8$ years (Range: $0-17$ years) in the entire survey sample, while it was $8.8$ years (Range: $0-17$ years) in those households with low SV and $6$ (Range: $0-16.3$ years) in those households with high SV. Additional comparisons between social vulnerability levels in terms of socio-demographic, health and socioeconomic information are demonstrated in Table 3. Households with high SV were often overcrowded, less educated, older, had a low number of income earners, had low levels of savings, and had less access to social security and health insurance compared to the low SV group. The statistically significant variable with the largest effect on social vulnerability was the average education of the household (Cohen's $d = 0.947$), followed by the ratio of income earners (Cohen's $d = 0.366$) and the ratio of over 65 years old in the household (Cohen's $r = 0.120$), having social security (Cramer's $V = 0.211$), having health security or insurance (Cramer's $V = 0.226$), having natural gas heating at home (Cramer's $V = 0.152$), the presence of anyone with a disability or who is elderly and needs care at home (Cramer's $V = 0.142$) and having savings for emergency situations (Cramer's $V = 0.135$).

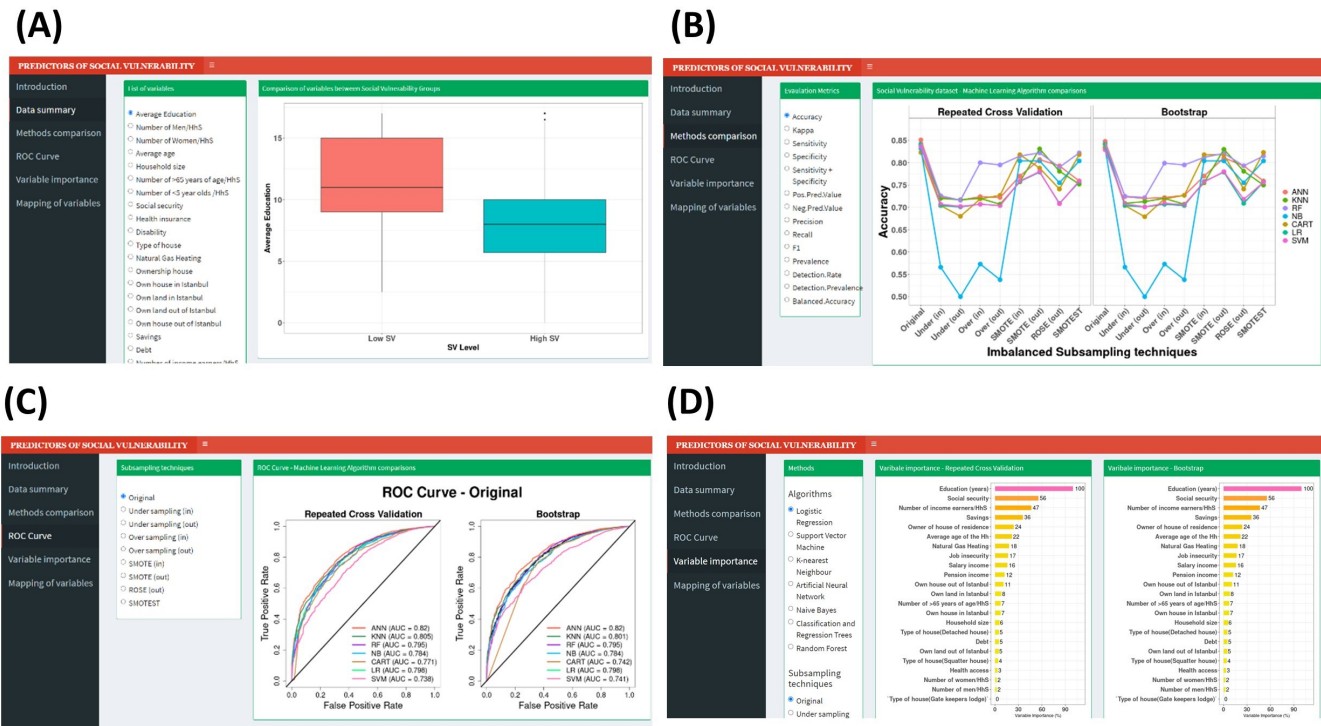

**Figure 2.** The components of open-access web application created in R Shiny interface. (can be accessed from https://oyakalaycioglu. shinyapps.io/Social_Vulnerability/). The left-side commands allow the user to choose which analysis to activate. (A) Summary statistics of the variables are visually compared across social vulnerability groups. Box plots and bar plots were used for continuous and categorical variables, respectively. (B) The performance metric is chosen by the user (y-axis) in comparison to the subsampling method (x-axis). The ML methods are displayed in different colours. Two separate plots are generated for RCV and bootstrap resampling techniques. (C) For the chosen subsampling method, LR and ML methods are compared in terms of the AUC of the ROC curve. Different coloured lines represent different methods. (D) For the chosen ML method and subsampling techniques, variable importance plots are displayed.

## 4.2 Comparison of machine learning methods

The comparison of the machine learning models in terms of their sensitivity, specificity, balanced accuracy, and AUC for different subsampling methods are presented in Fig. 3. The additional comparisons of models using other evaluation metrics (e.g., positive prediction value, negative prediction value, accuracy, F1 score, etc.) can be found in the R Shiny application Within these comparisons, no substantial differences were observed in the model performance indicators of LR and different ML strategies between RCV and bootstrap resampling methods. Therefore, we present the results that were obtained with repeated 5-fold cross-validation.

As mentioned earlier, the data set suffered from imbalanced class variables, particularly the outcome variable, and as such significant differences were observed when subsampling strategies were applied. Using the standard algorithm without sub-

sampling (referred as "Original") resulted in poor sensitivity (Fig. 3A), and inflated specificity (Fig. 3B) rates, due to the class imbalance in the studied sample where the negative class is dominant. Based on the criteria that $AUC > 0.7$, overall, the methods fitted with under subsampling inside the resampling procedure (referred as under(in)) performed better in terms of model performance metrics when compared to other subsampling methods. The highest balanced accuracy for each method was also obtained with under(in) subsampling (Fig. 3C).

In Table 4, all ML methods using under(in) subsampling were compared to their counterpart using the original data without imbalanced subsampling. Here we remind the reader that the priority in this study was to assess the performance of the models in terms of their success in identifying the households with high social vulnerability, which is the minority class, but therefore also the positive class. Using under(in) subsampling strategy demonstrated superior sensitivity and balanced accuracy rates compared to using original data and other subsampling strategies. Therefore, the results obtained with under(in) subsampling are considered for further comparisons between ML methods. Classification results for the ML models using under(in) subsampling are presented with ROC curves in Fig. 3D. The ROC curves for all other subsampling strategies with all other methods can be found in the R Shiny web application.

The best performing method in terms of AUC, accuracy, balanced accuracy, and sensitivity was artificial neural network using under(in) subsampling strategy (AUC: 0.813 (0.800-0.826), Accuracy: 0.724 (0.710-0.737), Balanced accuracy: 0.730 (0.790-0.752), Sensitivity: 0.740 (0.706-0.772), Specificity: 0.720 (0.705-0.735)). Naïve Bayes (NB) also produced a high sensitivity rate of 0.871 (0.843-0.894), however it resulted in significantly lower specificity (0.502 (0.485-0.519)) and overall accuracy 0.566 (0.550-0.581) compared to ANN (p=0.003 and p<0.001, respectively). While ANN balances sensitivity (0.740) and specificity (0.720), NB emphasizes sensitivity (0.871) over specificity (0.502). All other methods using under(in) sampling provided similar sensitivity rates between the range of 71.9% and 72.9%, and specificity rates between 69.9% and 72.4%. When AUC was considered, CART was also significantly worse than ANN (0.782 (0.768-0.796) vs. 0.813 (0.800-0.826), p = 0.005). Logistic regression, random forest, support vector machine and k-nearest neighbours did not show significant differences from ANN in terms of performance metrics.

Table 3: Univariate analysis of the study population characteristics.

| Variables | Social Vulnerability Level | | Effect size (Cohen's d[a] or Cohen's $r^b$ or Cramer's V[b] ) | P |
| | Low SV (n = 34,041) | High SV (n = 7,052) | | |
| --- | --- | --- | --- | --- |
| **Socio-Demographics** | | | | |
| Household Size (HhS) | | | $d = 0.178$ | $< 0.001$ |
| $mean \pm sd$ | $3.28 \pm 1.40$ | $3.54 \pm 1.72$ | | |
| $median(min-max)$ | $3(1-13)$ | $3(1-14)$ | | |

| | | | | |
|---|---|---|---|---|
| Average education (years) | | | $d = 0.947$ | $< 0.001$ |
| $mean \pm sd$ | $9.11 \pm 3.22$ | $6.11 \pm 2.9$ | | |
| $median(min - max)$ | $8.8(0 - 17)$ | $6(0 - 16.3)$ | | |
| Average age of the HH | | | $d = 0.107$ | $< 0.001$ |
| $mean \pm sd$ | $38.28 \pm 14.49$ | $39.87 \pm 16.65$ | | |
| $median(min - max)$ | $35.5(10.3 - 85.0)$ | $36.4(8.8 - 84.0)$ | | |
| No. of women / HhS | | | $d = 0.130$ | $< 0.001$ |
| $mean \pm sd$ | $0.48 \pm 0.23$ | $0.51 \pm 0.23$ | | |
| $median(min - max)$ | $0.5(0 - 1)$ | $0.5(0 - 1)$ | | |
| No. of men / HhS | | | $d = 0.130$ | $< 0.001$ |
| $mean \pm sd$ | $0.52 \pm 0.23$ | $0.49 \pm 0.23$ | | |
| $median(min - max)$ | $0.5(0 - 1)$ | $0.5(0 - 1)$ | | |
| No. of <5 years old children / HhS | | | $r = 0.130$ | $< 0.001$ |
| $mean \pm sd$ | $0.037 \pm 0.099$ | $0.039 \pm 0.088$ | | |
| $median(min - max)$ | $0(0 - 0.7)$ | $0(0 - 0.7)$ | | |
| No. of >65 years old individuals/HhS | | | $r = 0.120$ | $< 0.001$ |
| $mean \pm sd$ | $0.09 \pm 0.24$ | $0.15 \pm 0.30$ | | |
| $median(min - max)$ | $0(0 - 1)$ | $0(0 - 01)$ | | |
| Number of income earners / HhS | | | $d = 0.366$ | $< 0.001$ |
| $mean \pm sd$ | $0.53 \pm 0.28$ | $0.43 \pm 0.24$ | | |
| $median(min - max)$ | $0.5(0 - 2)$ | $0.3(0 - 2)$ | | |
| Social security, $n(\%)$ | $30956(90.9)$ | $5118(72.6)$ | $V = 0.211$ | $< 0.001$ |
| Membership to a non-governmental organisation, $n(\%)$ | $872(2.6)$ | $70(1.0)$ | $V = 0.040$ | $< 0.001$ |
| **Health** | | | | |
| Health insurance, $n(\%)$ | $33563(99.9)$ | $6206(88.0)$ | $V = 0.226$ | $< 0.001$ |
| Any disabled or elderly who needs care in the Hh, $n(\%)$ | $1112(3.3)$ | $789(11.2)$ | $V = 0.142$ | $< 0.001$ |
| Health access, $n(\%)$ | $28309(83.2)$ | $5682(80.6)$ | $V = 0.026$ | $< 0.001$ |
| **Socio-Economic** | | | | |
| Regular salary income, $n(\%)$ | $27342(80.3)$ | $4899(69.5)$ | $V = 0.100$ | $< 0.001$ |
| Pension income, $n(\%)$ | $11283(33.1)$ | $2320(32.9)$ | $V = 0.002$ | $0.668$ |
| Rent income, $n(\%)$ | $1794(5.3)$ | $180(2.6)$ | $V = 0.048$ | $< 0.001$ |
| Income support from public authorities, $n(\%)$ | $646(1.9)$ | $470(6.7)$ | $V = 0.111$ | $< 0.001$ |

| | | | | |
|---|---|---|---|---|
| Job insecurity in Hh, $n(\%)$ | 11808(34.7) | 2790(39.6) | $V = 0.038$ | $< 0.001$ |
| Ownership of the house of residence, $n(\%)$ | 22105(64.9) | 4057(57.5) | $V = 0.058$ | $< 0.001$ |
| Status of the house of residence, $n(\%)$ | | | $V = 0.087$ | $< 0.001$ |
| Apartment flat | 30453(89.5) | 5797(82.2) | | |
| Squatter house | 912(2.7) | 379(5.4) | | |
| Detached/semi-detached house | 2578(7.6) | 851(12.1) | | |
| Gate keepers lodge | 98(0.3) | 25(0.4) | | |
| Natural gas heating at home, $n(\%)$ | 31164(91.5) | 5580(79.1) | $V = 0.152$ | $< 0.001$ |
| Ownership of any other house in İstanbul, $n(\%)$ | 5667(16.6) | 585(8.3) | $V = 0.088$ | $< 0.001$ |
| Land ownership in İstanbul, $n(\%)$ | 2669(7.8) | 282(4.0) | $V = 0.056$ | $< 0.001$ |
| House ownership outside İstanbul, $n(\%)$ | 4210(12.4) | 491(7.0) | $V = 0.078$ | $< 0.001$ |
| Land ownership outside İstanbul, $n(\%)$ | 7092(20.8) | 889(12.6) | $V = 0.064$ | $< 0.001$ |
| Savings for emergency situation, $n(\%)$ | 5499(16.2) | 260(3.7) | $V = 0.135$ | $< 0.001$ |
| Any debt of Hh members, $n(\%)$ | 11009(32.3) | 2728(38.7) | $V = 0.051$ | $< 0.001$ |

[a] $0.2$ = a small effect, $0.5$ = a medium effect, $0.8$ = a large effect. [b] $0.1$ = a small effect, $0.3$ = a medium effect, $0.5$ = a large effect. HhS: Household size. No: Number.
Where Cohen's $d$ is given, independent samples t-tests is used; where Cohen's $r$ is given Mann-Whitney $U$ test is used; where Cramer's $V$ is given, Pearson's chi-square test is used.

### 4.3 Important predictors for the machine learning methods

In Fig. 4, a visual summary of variable importance analysis is presented as the relative importance of the predictors as indicated by the ML methods using under(in) sampling. As the methodologies used for analysing variable importance varies across different models, we averaged the variable importance rankings obtained with all models in Fig. 4A. The most important variable for every model is given a score of $100\%$, followed by the next important variable which takes a relative value between 0 and 100. The variables which appeared in top ten most influential variables in all seven models were education, having social security, the ratio of income earners in the household and having savings for emergency situations. Of these variables, the variable with the highest average importance was education.

In Fig. 4B we investigated the relative importance of the independent variables within the top-performing model, ANN-under(in), using the approach suggested by Garson (1991). Based on this model, the most important variable for the classification of households' social vulnerability appeared to be having social security. The other predictors with over $50\%$ of relative importance were a mixture of demographic and economic variables including living in a squatter house, job insecurity, ratio of the over 65-year-olds in the household, owning a house outside of İstanbul, household size, the ratio of income earners in the household and having savings for emergency situations.

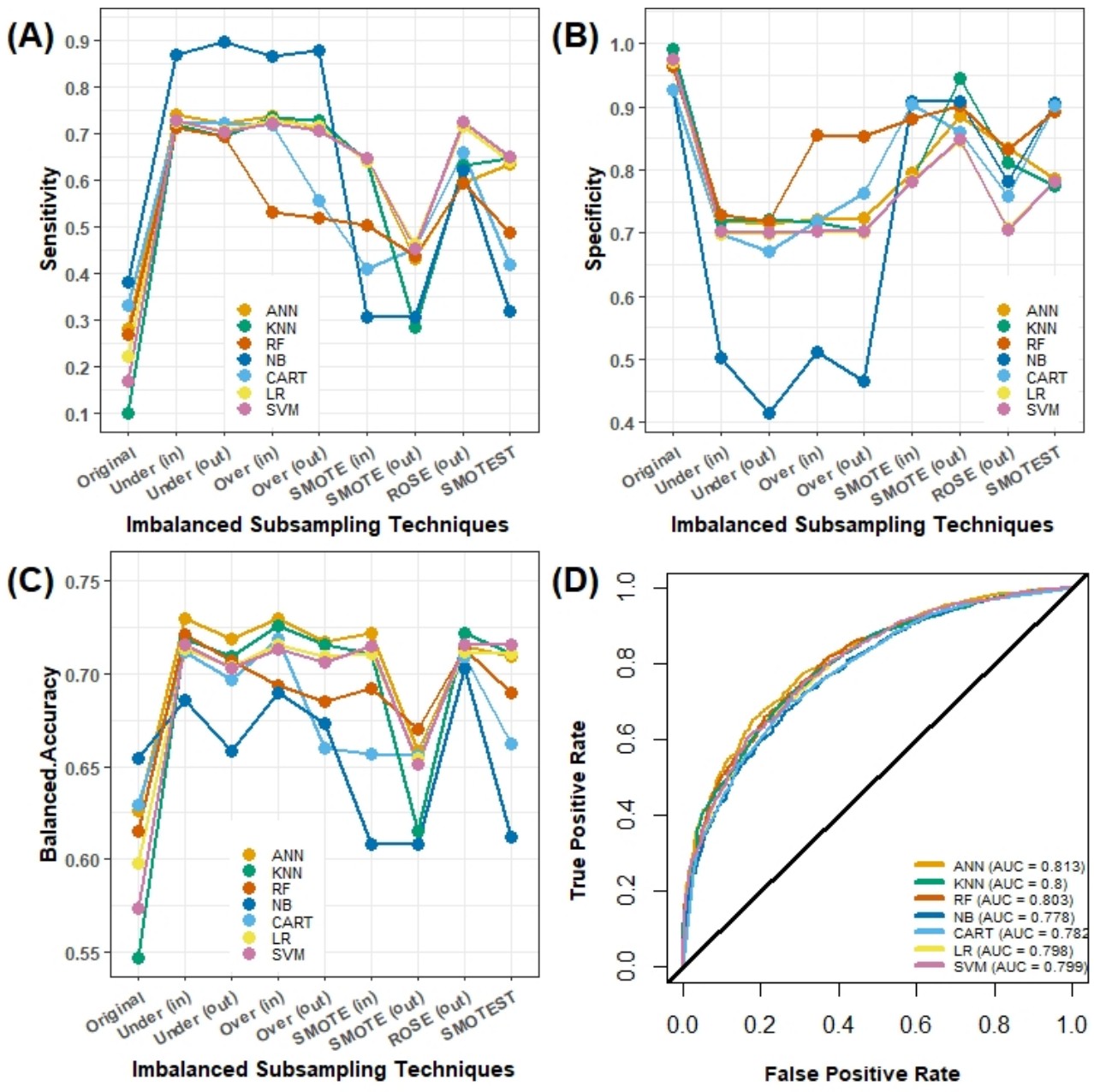

**Figure 3.** Model performance comparisons. LR and ML methods are visualized in different colours in all figures. (A) Sensitivity (y-axis) in comparison to subsampling technique (x-axis). (B) Specificity (x-axis) in comparison to subsampling technique (y-axis). (C) Balanced accuracy ((sensitivity + specificity) / 2) (x-axis) in comparison to subsampling technique (y-axis). (D) Using the under(in) imbalanced subsampling technique, ML methods are compared in terms of the AUC of the ROC curve.

**Table 4.** Comparison of the model performances of LR and ML methods using raw data and under(in) subsampling.

| ML Models | AUC (95% CI) | Accuracy (95% CI) | Balanced Accuracy (95% CI) | Sensitivity (95% CI) | Specificity (95% CI) | Diff sens* (95% CI) |
|---|---|---|---|---|---|---|
| *Original data (no subsampling)* | | | | | | |
| LR | 0.798 | 0.842 | 0.598 | 0.224 | 0.971 | NA |
| | (0.776-0.820) | (0.830-0.853) | (0.573-0.623) | (0.194-0.257) | (0.965-0.976) | |
| CART | 0.771 | 0.823 | 0.629 | 0.332 | 0.926 | NA |
| | (0.752-0.790) | (0.811-0.835) | (0.610-0.649) | (0.297-0.368) | (0.916-0.934) | |
| RF | 0.795 | 0.842 | 0.615 | 0.268 | 0.963 | NA |
| | (0.775-0.815) | (0.830-0.853) | (0.598-0.632) | (0.236-0.303) | (0.955-0.969) | |
| SVM | 0.738 | 0.836 | 0.573 | 0.170 | 0.976 | NA |
| | (0.709-0.767) | (0.825-0.848) | (0.560-0.586) | (0.144-0.200) | (0.970-0.981) | |
| NB | 0.784 | 0.832 | 0.654 | 0.382 | 0.926 | NA |
| | (0.767-0.801) | (0.820-0.843) | (0.635-0.673) | (0.346-0.419) | (0.917-0.935) | |
| K-NN | 0.805 | 0.838 | 0.547 | 0.102 | 0.992 | NA |
| | (0.772-0.838) | (0.826-0.849) | (0.535-0.559) | (0.081-0.127) | (0.989-0.995) | |
| ANN | 0.820 | 0.851 | 0.626 | 0.281 | 0.971 | NA |
| | (0.801-0.839) | (0.840-0.862) | (0.609-0.643) | (0.248-0.316) | (0.964-0.976) | |
| *Using under (in) subsampling* | | | | | | |
| LR | 0.798 | 0.704 | 0.713 | 0.726 | 0.699 | 0.502 |
| | (0.785-0.811) | (0.690-0.718) | (0.689-0.737) | (0.691-0.759) | (0.683-0.715) | (0.483-0.520) |
| CART | $0.782^{a}$ | 0.704 | 0.712 | 0.725 | 0.699 | 0.393 |
| | (0.768-0.796) | (0.690-718) | (0.690-0.734) | (0.690-0.757) | (0.684-0.715) | (0.373-0.413) |
| RF | 0.803 | 0.722 | 0.713 | 0.711 | 0.724 | 0.443 |
| | (0.790-0.816) | (0.708-736) | (0.692-0.734) | (0.676-0.744) | (0.709-0.738) | (0.421-0.465) |
| SVM | 0.799 | 0.707 | 0.715 | 0.72 | 0.702 | 0.559 |
| | (0.786-0.812) | (0.693-721) | (0.693-0.737) | (0.694-0.761) | (0.687-0.718) | (0.541-0.576) |
| NB | $0.778^{b}$ | $0.566^{a}$ | 0.690 | $0.871^{a}$ | $0.502^{a}$ | 0.489 |
| | (0.763-0.793) | (0.550-0.581 | (0.671-0.710) | (0.843-0.894) | (0.485-0.519) | (0.471-0.507) |
| K-NN | 0.800 | 0.720 | 0.719 | 0.719 | 0.720 | 0.617 |
| | (0.786-0.814) | (0.705-0.733) | (0.697-0.742) | (0.684-0.752) | (0.704-0.735) | (0.600-0.633) |
| ANN | $0.813^{a,b}$ | $0.724^{a}$ | 0.730 | $0.740^{a}$ | $0.720^{a}$ | 0.459 |
| | (0.800-0.826) | (0.710-0.737) | (0.709-0.752) | (0.706-0.772) | (0.705-0.735) | (0.440-0.478) |

*Diff sens: The difference in sensitivity between the same ML method with and without subsampling strategy for imbalanced problem. The same superscript letters indicate statistically significant difference in a performance measure between two methods, at $\alpha < 0.05/7 = 0.007$ significance level. CI: Confidence Interval. NA: Not Applicable.

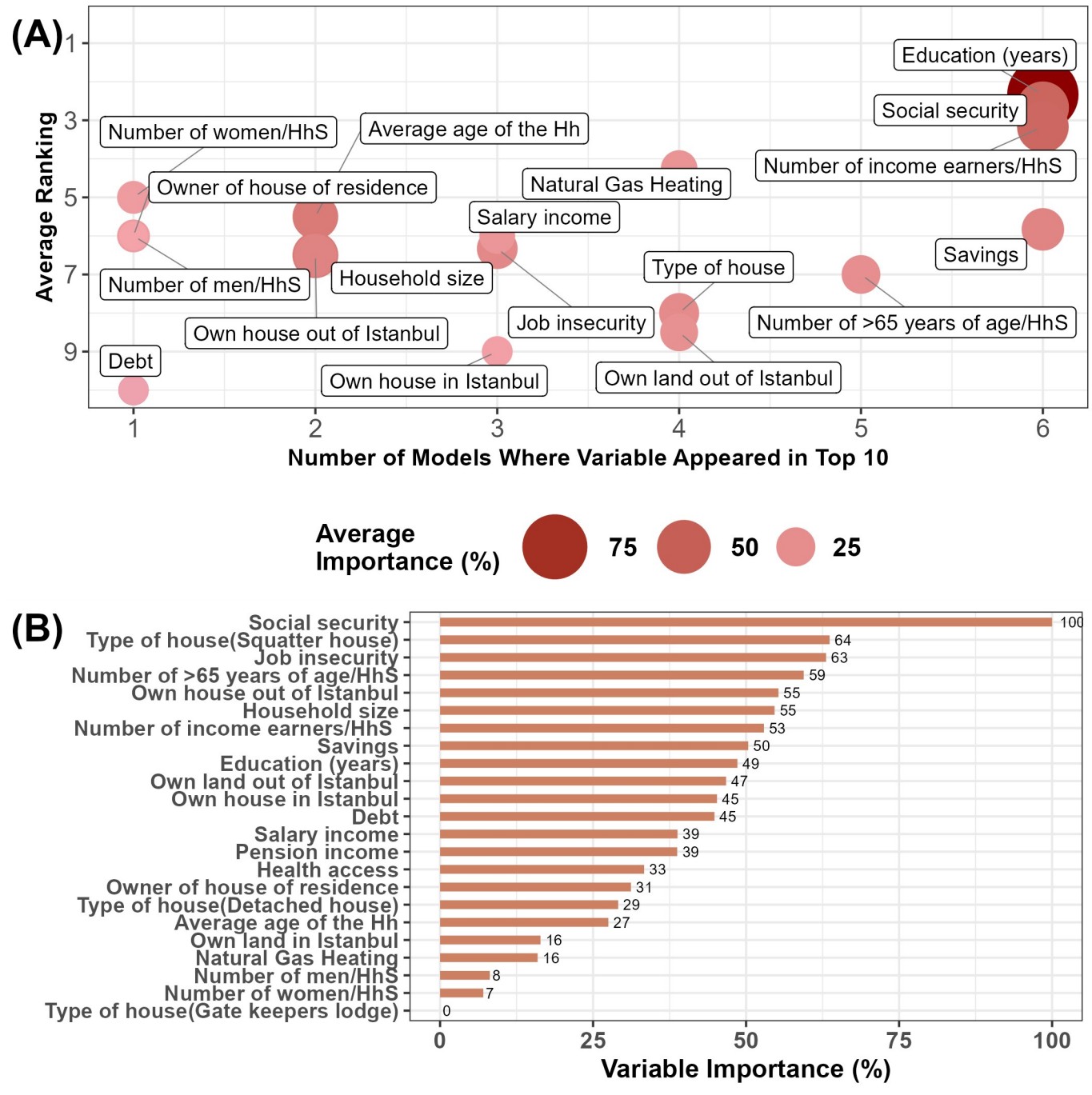

**Figure 4.** Important predictors for the assessment of social vulnerability. (A) the average relative importance of the predictors obtained with ML methods using under(in) sampling. Average ranking of the predictor across all models (y-axis) in comparison to a number of models that the predictor appeared in the top ten most important variables (x-axis). (B) Variable importance for the ANN-under(in) model.

### 4.4 Spatial distribution of the important predictors of the ANN model

Based on the variable importance analysis with the top-performing model, ANN-under(in), we performed area-based calculations to compare the neighbourhood characteristics in İstanbul. For categorical variables, the prevalence in the neighbourhood was calculated, while neighbourhood averages were used for the continuous variables. The three most important predictors of social vulnerability level were subsequently displayed as a five-category map in Fig. 5.

For Fig. 5A, the areas represented with dark red colours, below 70%, indicate those neighbourhoods with the lowest social security and these areas are prevalent in the outer regions of the metropolitan area. On the other hand, those neighbourhoods close to the central region mostly cover households with a higher prevalence of social security benefits. The number of neighbourhoods with high-density of squatter housing ($> 20\%$) was 27 (Fig. 5B). These neighbourhoods are scattered throughout the city and are not concentrated in any specific region. The households with job insecurity, are mainly located in the central region of the city (Fig. 5C). The distribution of all other variables across neighbourhoods of İstanbul can be found in the R Shiny web application.

## 5    Discussion

### 5.1    The selection of the optimal ML method

In this study, we demonstrated that it is possible to predict the social vulnerability of households with a certain degree of precision using household indicators available within the databases of various institutions and public authorities. Based on our results, the best-performing ML method for identifying households with high social vulnerability was ANN using under subsampling within the resampling procedure to address the problem of class imbalance (AUC = 0.813, balanced accuracy = 73%, sensitivity = 74%, specificity = 72%). ANN is often considered an effective and useful tool for identifying hidden relationships between socio-demographic and socio-economic variables that arise in social science research (Meade et al., 1970; Di Franco and Santurro, 2020). This may imply that, the interrelated social relations between the variables in our data set may be best handled by ANN. Apart from CART and NB, all methods provided similar AUC results (0.80) with no significant differences. There was no significant difference between the ML methods, except NB in terms of the performance of identifying households with high social vulnerability (i.e., sensitivity).

A model with an AUC greater than 0.80 was considered to have an excellent discriminative ability by Hosmer et al. (2013). Therefore, our proposed ANN model, with AUC of 0.813, indicated a good ability to discriminate households with high social vulnerability in a hazard event in İstanbul from those with low social vulnerability. Similarly, the AUC values achieved with RF and KNN were greater than 0.8. In terms of predictive accuracy, we obtained the largest balanced accuracy (73%) with ANN. Further, the accuracy obtained with ANN and other models did not differ significantly. We considered the accuracy of our optimal ANN model to be acceptable as the value is halfway between 50%, which is useless, and 100%, which is perfect (Power et al., 2013).

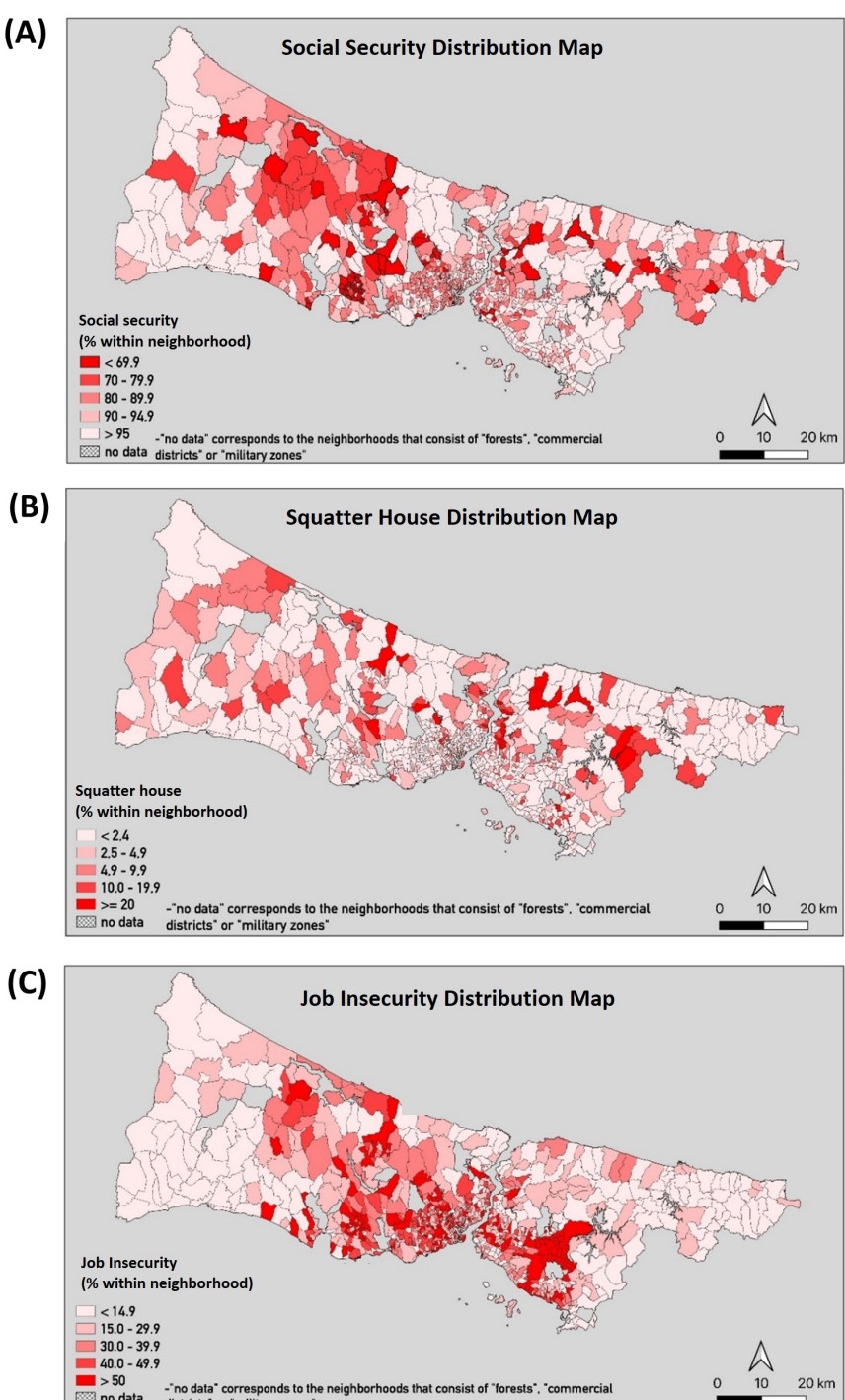

**Figure 5.** The five-category neighbourhood map of the three most important predictors of social vulnerability. (A) Neighbourhood prevalence of having social security (B) Neighbourhood prevalence of living in squatter houses (C) Neighbourhood prevalence of job insecurity of any household member.

A limited number of studies have used ML to predict hazard-related social vulnerability and reported performance metrics. Abarca-Alvarez et al. (2019) achieved an AUC of 0.780 using the CART model to predict the social vulnerability of residential units in Andulusia with dwelling variables. Similarly, we obtained an AUC of 0.782 with the CART model when under sampling was used. When demographic and social indicators were used with an ANN model, Abarca-Alvarez et al. (2019) obtained a balanced accuracy of 86.1%. Alizadeh et al. (2018) reported a high accuracy of 95.6% with ANN using regional indicators when predicting the social vulnerability of municipal zones in Tabriz, Iran. Compared to these studies we obtained a relatively lower accuracy with our ML models as we focused on proposing an optimal modelling strategy using readily available household variables. Thus, our modelling approach can be useful for decision-makers to take immediate action for the most vulnerable households and there is no doubt that the predictive performance of our models would benefit from incorporating more predictor variables.

## 5.2 The importance of subsampling for imbalanced class variables

An important aspect of our study was to find the most viable solution for the imbalance problem in our data set, as the imbalance ratio between the high and low SV groups was around 1/5. When no subsampling strategy was applied to handle imbalance problem, we obtained poor sensitivity rates. A 39.3% to 61.7% gain in sensitivity was achieved with different ML models when under(in) subsampling was applied, and therefore the imbalance was being addressed, compared to using the original raw data without subsampling.

In our study, when ML models without subsampling strategies were used, the overall accuracy was higher due to the inflated specificity compared to the models using subsampling strategies. The standard application of ML models targets to maximise the overall accuracy. Therefore, if they are trained on imbalanced data without considering imbalanced classes, they tend to over-predict the class with higher frequency (Esposito et al., 2021), which is the low vulnerability group in our data set. This increases specificity, and therefore reduces sensitivity. Therefore, the models based on the original imbalanced data resulted in lower sensitivity and failed to identify households with high social vulnerability, and they failed to meet our aims in the study.

Among subsampling methods, the random majority under sampling approach resulted in the best performance for all ML methods. This method discards data points from the majority class (i.e. low vulnerability group) at random until a more balanced distribution is reached, while training models. Our data set was sufficiently large not to be negatively affected by the discarding of data. Our results obtained with random under sampling are consistent with the ML literature, in the sense that if the size of the data set is large, then it is better to employ an under sampling method (Durahim, 2016).

## 5.3 Important variables and their theoretical implications

Variable importance rankings tended to differ depending on the technique employed. Therefore, initially we aggregated the results of the variable importance analysis. On average, education was found to be the most important variable in all methods, followed by having social security, the ratio of income earners in the household, and having savings to be used in emergency situations. Within the top performing model, ANN, the most important variable was found to be social security, followed by living in a squatter house and job insecurity. When we discuss these results based on socio-urban conditions in Türkiye, we

can easily comprehend that education and social security are interrelated factors as more educated citizens tend to work in jobs with social security. Second, income and savings represent households' economic power to cope with hazards.

Social security refers to the right to have the guarantee of unemployment benefits, retirement pensions, public protection from job injuries, and access to public health coverage, gained through regular work and employment (The Republic of Türkiye Ministry of Labour and Security 2021). The lack of social security and insurance, particularly in a demonstrably unstable economy, increases vulnerability to many kinds of crises, including disasters and health emergencies such as pandemics. In our research, having social security actually means being able to get different kinds of socio-economic and health support in sudden shocks, which also cover the aftermath of a hazard as the individual is registered in the public health system. In Türkiye, the rate of unregistered labourers who are not affiliated with the Social Security Institution in total employment was $27.4\%$ (Turkish Statistics Institute, 2021), while most unregistered labourers were found in the agriculture and service sectors (Ocal and Senel, 2021). Unregistered employment means that no social insurance premiums are paid by the employer; thus employees cannot have the benefit from social security (Turkoglu, 2013). However, people in agriculture are mostly self-employed and do not have social security because they cannot afford to pay social security premiums regularly. Hence, the map we have presented on the different social security status of neighbourhoods with respect to the household survey indicates the northwest of İstanbul with lower social security, which may be due to a large amount of agricultural areas in that region. However, those neighbourhoods close to the centre of the İstanbul metropolitan area are mostly inhabited by people employed in the services and industrial sectors, with a higher rate of registered employment and thus a higher prevalence of social security benefits. Moreover, in the data presented, the prevalence of social security in the high vulnerability group is around $72\%$, whereas it is as high as $91\%$ in the households with low vulnerability.

Based on our findings, living in a squatter house was the second most important variable of social vulnerability using the ANN method. Squatter housing comprises houses that are assembled quickly and do not conform to the technical and legal standards (called "gecekondu" as the Turkish name for poor squatter settlements). Hence, this type of housing represents at-high-risk buildings in the event of geological and climatic hazards and is more likely to be damaged in such events which implies higher vulnerability to hazards. One of the large-scale hazardous events anticipated for İstanbul is an earthquake with a magnitude greater than 7 Mw, which is predicted to strike the city within the next 30 years with $42 - 47\%$ probability (Murru et al., 2016). Previous studies inform that a large proportion of buildings in İstanbul, including squatter settlements, are not earthquake-resistant (IMM and KOERI, 2019; Parsons, 2004; Ersoy and Koçak, 2016; Erdik et al., 2003; Atun and Menoni, 2014). Furthermore, squatter housing is linked to poor socio-economic household profile. It is known that poorer people are more vulnerable to natural hazards as they settle in buildings at higher risk but more affordable to them because of cheap rents (Salami et al., 2015). In particular, squatter houses are very low-quality buildings, and when taken together with the poor socio-economic characteristics of their residents, they represent high social vulnerability for households. A study by Abarca-Alvarez et al. (2019) in Andalusia, which used CART, showed the importance of dwelling variables on social vulnerability, such as the average age of constructions and the density of buildings in a particular district of an urban area. In our study, the age of the buildings was not available in the data, however, the type of housing was found as an important predictor of social vulnerability.

With the ANN method, the third highest-ranked variable was job insecurity. The spatial distribution of neighbourhoods in terms of job insecurity indicates that the centre of İstanbul close to the Marmara Sea is densely populated with households with job insecurity representing the possible unemployment figures in those crowded areas. Further, as mentioned above in the social security indicator, the labour market opportunities in Türkiye are highly dominated by the casual or seasonal employment opportunities (Ocal and Senel, 2021). Such forms of casual employment are highly fragile since the labourers are not in full employment and not registered in the social insurance system. A recent study showed that casual and unregistered employment increases social vulnerability to natural hazards (Mavhura and Manyangadze, 2021). These may be either in the form of casual, seasonal employment or self-employment, where social security and social insurance registrations are not provided by the employers and the employees could not afford to pay their premiums regularly by themselves. These types of employees and small businesses mostly fall below the poverty line even if they may be observed as working (Adaman et al., 2015). Those households which depend on casual, unregistered employment and small businesses have a high probability of experiencing vulnerability when a disaster strikes as they may experience loss of any economic means in that situation. There is an important difference between job insecurity and social security variables. Job insecurity actually reflects the situation where the individual has no regular income, on the other hand, social security is covering all kinds of support and compensation mechanisms not only limited to the economic means of regular income. Although not limited to these, there might be several reasons for the difference between neighbourhoods in terms of these two variables. For example, it may be that in the rural areas of northwest İstanbul, the individuals may not have social security, but they own their land and small businesses and their jobs are more secure even though they may have a limited income (Acar et al., 2022). In contrast, in the centre of the city most of the population is in wage employment where a major group is in regular registered employment besides a significant group of unemployed or those working on a daily basis in casual jobs (Acar et al., 2022). Hence, unemployed or those in daily jobs may suffer job insecurity and high risk of losing employment and/or income if caught by a hazard. Moreover, the individuals working in the service sector, which is common in İstanbul neighbourhoods, may suffer more from the possibility of work closures after a major hazard. For example, during the COVID-19 pandemic, when small workplaces have been required to close or restrict their services for a long period of time, most working people suffered severe job and income losses, hence high vulnerability emerged (Bartik et al., 2020; Gray et al., 2022). While İstanbul took $41.9\%$ share of the total services sector in Türkiye in 2021, the share of the services sector in İstanbul's total gross domestic product was $33.7\%$ (Turkish Statistics Institute, 2021).

The other variables among the top ten most important predictors that contribute to the model performance of the ANN model were a mixture of demographic and economic variables. These included the ratio of over 65-year-olds in the household, owning a house outside of İstanbul, household size, the ratio of income earners in the household, having savings for emergency situations, owning land outside of İstanbul, and the level of education of the residents. The demographic variable of having elderly ($> 65$ years) in the household being an important predictor of social vulnerability to hazards is also highlighted in the literature (Chou et al., 2004; Fatemi et al., 2017). High education which lowers social vulnerability is a factor that is both related to having social security, as mentioned before, and with an increase of awareness to take precautions for possible hazards. The other significant variables like having property and savings are both related with income, where the property outside the city

may give more chances for the households to have a safe shelter after a major hazard. Furthermore, the associations between income and level of education are strong and consistent; that is children from poorer family backgrounds have a tendency of achieving a lower level of education (West, 2007). Also, the poor have less access to resources which may be effective in reducing risks, such as extra savings for preparing their houses to a hazard or accessing risk preparation information, and therefore cannot take as many precautions to cope with a disaster when it occurs (Hallegatte et al., 2020).

## 6   Limitations and recommendations

Socially, economically, and environmentally vulnerable communities are more likely to suffer disproportionately from disasters (Cureton, 2011; Hallegatte et al., 2020). However, our analysis was based solely on quantifiable household data, since variables related to environmental factors, historical hazard data and building infrastructures were not available in our survey-based data set. Another important limitation regards the fact that we are using social vulnerability index scores that are pre-constructed in a previous social vulnerability research. As we aim to assist the social vulnerability assessment process of local authorities, which is IMM in our case, we do not tend to discuss their scoring scheme as it is part of their official policy-making process, but we try to present them a methodological approach based on machine learning techniques to identify the best possible predictors of social vulnerability. However, as urban growth and migration are common experiences in a vibrant city like İstanbul, by regeneration and renewal processes accelerating the trend, the location of residents is continuously changing similar to the change in socio-economic positions of neighbourhoods both upward and downward. This may result in a continuous change of status and dynamic social vulnerability of households and neighbourhoods which needs to be studied in further research.

Although assessing social vulnerability is a complex process that takes many personal and environmental factors into account, our predictors in the ML models were limited to quantifiable household data as our aim in this paper is to present an optimal modelling strategy capable of processing readily available large databases. Therefore, the model accuracy with the final ANN model was relatively low compared to other studies which assessed social vulnerability to hazards with machine learning techniques. For future studies, we recommend using household data along with community-level spatial predictors to enhance the predictive ability of the models. We note that we could not perform an external validation of the ML models using an independent data set due to the unavailability of such household data derived from another source. Although the models were tested using independent testing data from our survey data, the model predictions may benefit from validation studies which could be conducted using independent data sets.

## 7   Conclusions

This research presents a new and alternative approach for public authorities to develop ideas for future governance mechanisms to cope with social vulnerability based on interdisciplinarity as a combination of social and statistical science. To address the social vulnerability predictors by using ML, we compared six different supervised machine learning techniques and logistic regression which can be employed for binary classification with imbalanced class variables. We demonstrated that an ANN

using majority under sampling was the optimum method in terms of sensitivity, AUC, and other relevant performance metrics. The variable importance results showed that economically deprived households which do not have social security and experience job insecurity, the ones living in squatter houses and less educated individuals are more likely to have a high social vulnerability to hazards. We stress strongly that our research outcomes and demonstration of employing machine learning with large household-level data have the potential to support decision-makers to develop more effective policies by making use of quantifiable household data which are available across various institutions and public bodies. More explicitly, a policy-maker can make use of our proposed final ANN model to discriminate between households with low and high social vulnerability, by inputting the variables found significantly important in the study. Thus, the groups with certain characteristics which are more vulnerable may be prioritised by decision-makers in terms of their needs in order to develop new schemes that are specifically targeted for reducing disaster related vulnerabilities. This kind of targeted assistance is missing in Türkiye's local and national disaster risk reduction policies, though it is a part of the Sendai Framework (UNISDR, 2015). Therefore, the local authorities, mainly Municipalities, can benefit from the results of this study to target poor groups to accommodate them in affordable disaster-resistant housing within urban renewal schemes, for improving social assistance for the elderly, children, youth, and the poor, and for increasing awareness-raising events. Also, the central authorities may define new policies for increasing access to education and to social security of the poor and the vulnerable groups. This study made use of machine learning methodology and assessed their performances on social data based on an interdisciplinary collaboration where the statistics, urban planning and sociology disciplines intersect, to understand the significance of assessing social vulnerability at the household level and how to build a society more resilient to disasters.

**Supplementary Materials:**

Supplementary File 1: Data Source: Social vulnerability research and construction of SoVI in 2017

Supplementary File 2: Data sources from which the predictors of ML models can be obtained

Supplementary File 3: Machine Learning Methods

*Code availability.* R codes can be obtained by contacting Oya Kalaycioglu at her e-mail address: oyakalaycioglu@ibu.edu.tr.

*Data availability.* Data are available from the authors with the permission of İstanbul Metropolitan Municipality, Directorate of Earthquake and Ground Research.

*Author contributions.* OK and EYM planned the initial concept of the study. OK led the writing of the manuscript with contributions from all the co-authors. OK and SEA implemented the data analysis, trained ML models, and designed tables, figures and R Shiny web

application. EYM obtained the data and designed Fig. 5. MK and SK wrote literature review on social vulnerability and discussion on

important predictors. All authors critically reviewed the manuscript.

*Competing interests.* The authors declare that they have no conflict of interest.

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
