# Peer review of "Using machine learning algorithms to identify predictors of social vulnerability in the event of a hazard: İstanbul case study"

_Natural Hazards and Earth System Sciences, 2022_

## Author Response (AR1)

Dear Editor,

First of all, we would like to take to opportunity to thank you and the reviewers for valuable and insightful comments. We appreciate the time and effort that you have dedicated to providing feedback which improved our proposed manuscript to a great extent. We believe that, in the revised version we clarified the misunderstandings regarding our aim by providing more details.

Here we note that, we were over ambitious in the previous manuscript to state that our model helps to reduce disaster risk. In the revised version, we made it clear that by using ML methods our aim is to propose an approach which can help in the decision-making processes for policy-makers to assess "who is socially vulnerable?". In this context, we assessed what are the variables that contribute household-level vulnerability the most. By using our proposed ML model, a policy-maker can input these variables – which are easily accessible by policy-makers – to our proposed final ANN model and discriminate the households with low and high social vulnerability (SV) as well as identifying which indicators make which households most vulnerable since the vulnerability of households are different. Hence our model proposal has potential to reduce the time and economic burden that one may spend to conduct surveys to predict household level SV. Therefore, our proposed model is not directly focused on reducing disaster risk, but it would help policy-makers to facilitate risk reduction activities targeted for vulnerable groups. Also, we want to clarify that we did not intend to calculate social vulnerability index (SoVI) in this study. We used pre-constructed SoVI as an outcome in the present study, which was the only SV measure available for Istanbul.

We hope that our revision will satisfy the proposed requirements for successful publication. Please see below for a point-by-point response to your comments and concerns.

Yours sincerely,
All authors

**Response to Editor's comments:**

1. Providing an improved description, literature review, and discussion on social vulnerability indices, methods, and theory. Both reviewers pointed out several areas of research that are not acknowledged in the manuscript, including empirical modeling for social vulnerability, critiques of social vulnerability indices, the underlying theory for social vulnerability, and social science research in the event of an earthquake, among many others. The authors should engage with this literature throughout the manuscript—in the introduction, methods, results, and discussion. This also includes using the correct terminology for social vulnerability research and avoiding phrasing that contradicts basic concepts of social vulnerability.

**Authors' response:** In the revised manuscript we improved literature and discussion regarding social vulnerability in hazards. We have added a *Section 2* which is devoted to determinants of social vulnerability, giving examples from social science research and theory of social vulnerability, summary of approaches to measure social vulnerability indices and empirical modelling of social vulnerability. Considering both reviewers' suggestions on terminology, we have corrected the terminology throughout the manuscript, assets, and the supplementary files in accordance with the vulnerability literature.

2. Addressing comments on the proposed framework to predict SVI (an index) with household-level social variables. Both reviewers would like to read justification of this approach. This broadly requires a thorough description of using SVI as the predicted outcome (as opposed to other measures), and ensuring that SVI is not equated with disaster impact/losses/recovery. The selection of predictor variables should be described, including more detail on why/how they were selected, how they were developed, and whether they'd be available at the household level in order to apply this approach in the future. A practical comparison between this approach and the traditional SVI approach in Istanbul should also be completed. Methodologically, this includes a detailed description of the construction of the SVI score that was used as an outcome since the documentation is not easily accessible and addressing concerns about using this as a binary outcome.

**Authors' response:** In the revised version, we have included a new section (*section 3.2.2)* to explain how SoVI was constructed in the previous study in 2017. We have also translated this work conducted by Istanbul Metropolitan Municipality to English and included it as a *supplementary file 1.* We are aware that, in the literature there is a great variety of indicators

that are being used to assess social vulnerability, as social vulnerability itself has a complex nature. In IMM's study, SoVI was calculated following the Cutter et al.'s (2003) factor analysis framework and 53 indicators, which were reduced to 7 factors. Thus, in the construction of SoVI, social vulnerability was considered as a capacity of society and individuals to cope with hazard and damage. The indicators chosen for the calculation of SoVI have been selected following extensive literature reviews and discussions with experts (please see *supplementary file* 1). Additionally, SoVI calculated by the previous study by IMM is taken as the outcome of ML models by the present study and to verify the results or validity of it is not among the aims of the present study as made clear in *section 3, paragraph 2*.

Here we note that, the "physical" components of risk were out of the scope of our study and we focused on social aspects of vulnerability. Currently, SoVI developed by IMM is the only theoretically grounded measure for SV in the context of Istanbul. Thus, we used it as an indication of social vulnerability of the households in Istanbul. Here we note that it is quite challenging to access/find quality empirical information regarding disaster-related topics, particularly in Turkey as in many developing countries and the Global South context. Information related to historical data on disaster impact/losses/recovery is mostly not in place for smaller regional units in Turkey, then even if it is there (gathered by related institutions), it is not shared. Therefore, as empirical data is not in place, index-based methodology to represent social vulnerability was based on the Cutter's factor analytic framework (Cutter et al, 2003) in relation to earthquake hazards was used in the study conducted by IMM, as in many developing countries. Please see *Section 3.2.2* and *Supplementary File 1* for more details.

For the current study, we aimed to build a ML based model, that can predict the SV level of the households in Istanbul using quantifiable predictors which can be obtained from various institutional databases without requiring a household-based survey. These predictors are related to socio-demography and socio-economy of the households, and housing. Those information are available from Municipalities, Address Based Population Registration System, Social Security Institution, National Taxation Department, Land Registry and Cadastre, Ministry of Health, Ministry of Family and Social Services, and Banks. The institutions with which the variables used in the study can be obtained are given in the *Supplementary File 2*. Due to confidentiality of household level data, these variables would not be available publicly, however a policy-maker can easily access these data from these institutions.

In the revised manuscript, we have also added a paragraph to clarify how our SoVI prediction and model differs from the previous work on SoVI construction. For details, please see *Section 1, paragraph 3*. We also justified why we have used SV as a binary outcome in *Section 3.3*.

**3.** Addressing concerns about whether this approach is specific to earthquakes.

**Authors' response:** The pre-constructed SVI which was used as the dependent variable in our present study, was calculated focusing on earthquakes. However, the indicators that we used for predicting SV level in the present study can be implemented more generally to assess the possible impact of social vulnerability due to other types of hazards, as made clear by both reviewers. Therefore, we do not intend to claim that selected variables are specific to earthquake vulnerability, and our model can be applied to any major hazard that can affect macro (i.e. roads, bridges, public buildings, etc.) and micro (i.e. housing) infrastructure. We made this clear in *Section 3.4, lines 263-265*.

**4.** Clarifying, in a thorough manner, how this approach could be implemented by stakeholders. This includes clearly stating who those stakeholders are, how they would apply the ML approach proposed in this manuscript, and a potential use case.

**Authors' response:** Our motivation was to assess the variables which contribute most to the social vulnerability of the households, given a pre-developed SVI as an outcome. By using our proposed ML model, a policy-maker can input these variables to our proposed final ANN model and obtain the status of social vulnerability of the household. Therefore, the groups with the certain characteristics which are more vulnerable may be prioritised by decision makers in terms of their needs in order to develop new social assistance schemes that are specifically targeted to disaster vulnerability. Such kind of targeted assistance is missing in the local and national disaster risk reduction policies in Turkey, though it is a part of the Sendai Framework. The stakeholders are but not limited to metropolitan / district municipalities, city governorship and public authorities. Hence our model proposal has potential to reduce the time and economic burden that a decision-maker may spend to conduct surveys to calculate household level SVI. We clarified the contribution of using ML methodology in relation to predict SV in the revised manuscript. We have included these details in *Section 7, lines 654-665* in the revised manuscript.

**Responses to Reviewer 1's comments**

As an open reviewer of this manuscript, I first thank the handling editor of this special issue of NHESS, for giving me the opportunity to conduct the review. Next, let me introduce myself to demonstrate my qualifications for this review. My name is Yi Victor Wang (https://dryvw.com/). I am currently serving as a Postdoctoral Fellow at the Institute for Earth, Computing, Human and Observing (ECHO) at Chapman University, Orange, California, USA. I have been authentically studying and researching in the scholarly field of science, engineering, and management of hazards and disaster risks for over a decade. I have a bachelor's degree, Master's degree, and a Ph.D. degree in this field. One of my major academic contributions so far is the proposal of an *empirical predictive modeling* approach to quantifying disaster vulnerability with consideration of social factors (a.k.a., *social vulnerability*). To facilitate communications regarding this review on the topics of social vulnerability to natural hazards, especially in the event of an earthquake, from my perspective, I recommend that the authors take a look at my first-authored peer-reviewed journal papers of Wang et al. 2019, 2020, and 2021 as well as Wang and Sebastian 2021 listed at the end of this review. In particular, Wang et al. 2021 is highly pertinent to what has been covered by the authors' manuscript.

**Authors' Response:** We thank the first reviewer Dr. Wang for his valuable comments on our proposed manuscript. Please see below for a point-by-point response to his comments and concerns.

**General Comments**

In terms of the authors' manuscript and research, I like the idea of applying machine learning (ML) methods to quantify social vulnerability and to identify predictors of social vulnerability. I also appreciate the technical prowess of the authors manifested in their statistical analyses. Having said these, however, I believe that the current version of the submission is far from the level of acceptance for publication. There are a number of major issues that render the manuscript highly dubious. The story line is also logically unsound and broken at several locations of the manuscript. The way the manuscript is laid out exposes the authors' lack of knowledge, confidence, and familiarity in topics related to disaster vulnerability and natural hazards. The authors have spent a disproportionately large amount of effort in showing the technical details of a few selected sections of their research that are actually not highly important regarding the purposes of their research. The motivations, results, and discussions in the manuscript around the topics, that are supposed to be pertinent to the practices to improve earthquake disaster risk reductions, are presented in a highly superficial manner. In order to receive a green light from me, the authors need to solve the major and minor issues as listed below and conduct a thorough revision to their manuscript accordingly in the later stage of the review process.

**Authors' Response:** WE thank for appraising the technical strengths of our paper. We revised the manuscript accordingly in the new version, by improving introduction, literature review, and discussion on social vulnerability indices, methods, and theory as made clear by our responses below.

**Specific Comments**

1.      L1: The uncountable noun of vulnerability in disaster research, especially for risk assessment for prediction of future loss, essentially means the propensity of an entity towards loss given a unit exposed value (such as life, economy, health, livelihood, infrastructural functionality, etc.) when the entity has experienced a certain level of hazard strength (such as ground shaking of an earthquake, wind gust of a tornado, inundation of a flood, etc.). In addition, vulnerability is usually also considered to be associated with the tendency towards a long-term suffering due to poor recovery by many, especially social scientists. To facilitate the management of disaster vulnerability before an unwanted event occurs, we may conceptualize disaster vulnerability as a combination of social vulnerability due to social factors, environmental vulnerability due to environmental factors, infrastructural vulnerability due to infrastructural factors, etc., as described in many classical literatures such as Cutter 1996 (https://doi.org/10.1177/030913259602000407). By the way, this Cutter 1996 is not the paper cited in the authors' manuscript. In the early days without big data on reliable and sufficient historical records of disaster losses, practitioners and scholars needed some method to quickly estimate disaster vulnerability. When it came to social vulnerability, professionals found that using social factors to construct a social vulnerability index (SVI) seemed to be a good approach for measuring social vulnerability. However, SVI itself is not social vulnerability. It is an indicator/predictor of social vulnerability at most. In the title, the authors claim that their research was to identify predictors of social vulnerability. But according to the body of the manuscript, it is clear that what the authors actually did was to identify predictors of an SVI. This is equivalent to building models to establish the relationships between a set of social variables and another set of social variables. What is the point for doing this when the authors could simply add these so-called predictors directly into their SVI?

**Authors' Response:** In the revised manuscript, we have completely revised the Introduction and added a new section (Section 2) to give more information on the concept of hazard related social vulnerability. We have updated the terminology in accordance to the SV literature. As the reviewer made clear in his comment, we used pre-constructed SoVI as an indication of social vulnerability. In the revised version we clarified that "For building ML models, we rely on a large dataset of a previous study comprising a household survey (n=41,093) and pre-constructed social vulnerability index (SoVI) of these households. We consider the SoVI scores as an indication of the social vulnerability level for each household, and our focus in this study is to assess to what extent the pre-constructed SoVI (and hence the social vulnerability of the households) can be predicted with machine learning techniques using household data that are available within databases of various institutions and public authorities.". Please see *section 1, paragraphs 3 and 4* for concise explanation of the aim and contribution scope of the study, when compared to previous study where SoVI was obtained.

2.   Then, regarding the SVI in the authors' research, I am not sure how the authors could resolve this second issue satisfactorily. As I have said in the previous comment, the original efforts to create SVIs were limited by a lack of sufficient historical records of event losses. Now, we are in year 2022 in the age of big data. We are having access to a gigantic amount of historical records of event losses to support empirical modeling of disaster vulnerability, socially, environmentally, infrastructurally, or in whatever manner. Why do we have to get stuck with the non-empirically derived SVIs to guide disaster risk reduction practices? For those SVIs that cannot be verified with historical data on losses, they are not reliable for offering any policy suggestions. For those SVIs that can potentially be verified with historical data on losses, it would be more appropriate to directly establish empirical models of disaster vulnerability

with calibrations of models on the historical data on losses. Without empirical evidence that directly associates with the expected event losses or poor recovery processes, any SVI is merely a product of social construction based on amplified voices from a seemingly scholarly, but actually perhaps more political than academic, echo chamber that eventually result in the production of some form of emperor's new clothes more or less.

**Author' response:**

In the revised manuscript we discussed how the prediction of social vulnerability would be enhanced by empirical modelling based on historical event data and intensity measures for the given hazard in addition to theoretical indicators of social vulnerability (please see *Section 2, lines 144-154*). Unfortunately, it is challenging to access/find quality empirical information regarding disaster-related topics, particularly in Turkey as in many developing countries. We added a paragraph explaining this limitation in *Section 3.2.2, lines 238-242*. We are aware of the open data sources and statistics available but none of them was sufficient for making decisions in a complex urban environment such as Istanbul. Such aggregated data sources include too many assumptions and uncertainties that hinder risk informed decisions at the local level. We agree with the value of empirical research but we had to rely on pre-constructed SoVI which is based on survey research and literature reviews and secondary sources to identify social vulnerability predictors. In *section 3.2* in the revised version and *supplementary file 1*, we provided more details on the previous "social vulnerability research" conducted in Istanbul and how SoVI was constructed. To date, this work by Istanbul Metropolitan Municipality is the most comprehensive and theoretically grounded study for assessing social vulnerability of households in the event of a hazard, which was originally constructed for earthquake as a most probable major hazard risk for Istanbul, Turkey. In this paper we do not intend to discuss the SoVI scores or the methodology of this previous study, as we would like to recommend an ML based modelling approach to decision-makers that can assist their current policy-making context as we made clear in *Section 3, lines 186-200.*

It is also worth highlighting that this study is not based on "physical" components of risk but rather "social". Therefore, we focused on social characteristics of a given context rather than physical ones.

3.       L2: The title emphasizes "social vulnerability in the event of an earthquake". While reading the manuscript, however, I could hardly find anything to support the hypothesis that the manuscript is about vulnerability to an earthquake event. The input variables of the ML models have nothing to do with earthquakes. The authors have also failed to show why the output variables of the ML models are for an earthquake event. It seems that the data of the research is based on a survey by the Directorate of Earthquake and Ground Research of Istanbul Metropolitan Municipality. Although the name of this organization involves earthquake, the variables of the survey used by the authors seem to be totally unrelated to earthquake events. There are no measures of hazard strengths of earthquake events, such as local magnitude, moment magnitude, peak ground acceleration, peak ground velocity, peak ground displacement, peak spectral acceleration, modified Mercalli intensity, etc. The authors need to justify why their work is for an earthquake event or for earthquake events.

**Authors' response:** The analysis presented in this research is based on a survey data, which was carried out by the "the Directorate of Earthquake and Ground Research of Istanbul Metropolitan Municipality". The data available were collected through face-to-face interviews conducted with the selected households. The interview questions used to derive SoVI include

questions related to respondents', housing conditions, preparedness to earthquake, their risk of perception to earthquakes and their past experience on earthquake incidence (please see *supplementary file 1*). Although the SoVI was focused on earthquakes in the previous study, the indicators that we used for assessing SV level in the present study can be implemented more generally to assess the possible impact of social vulnerability due to other types of hazards. Therefore, we do not intend to claim that selected variables are specific to earthquake vulnerability, and our model can be applied to any major hazard that can affect macro (i.e. roads, bridges, public buildings, etc.) and micro (i.e. housing) infrastructure. We made this clear in *section 3.4, lines 263-265*. We have also updated the title of our manuscript in this context. As explained above in our response number 2, we focused on as our context was social vulnerability rather than physical.

4.      L27: The term of "social vulnerability risk" or "risk of social vulnerability" that also appears later in the manuscript is exceptionally confusing. As I have referred to the meaning of social vulnerability previously and the word "risk" also has its specific meanings, what is the meaning of this "social vulnerability risk"? For a summary of the meanings associated with the word "risk" in scholarly works, the authors may have a look at Möller 2012 (https://doi.org//10.1007/978-94-007-1433-5_3). It seems that, with their survey data, the authors created two categories, i.e., a high SVI and a low SVI, based on a cutoff score. So, why do the authors have to call these two categories "severe risk of social vulnerability" and "non-severe risk of social vulnerability", instead of "high SVI" and "low SVI"?

**Authors' response:** We agree with the reviewer that the terminology that we used for the SV groups in the previous version was confusing. Accordingly, in the revised the manuscript we refer the groups of social vulnerability as high SV and low SV. We have updated all the supplementary files and assets accordingly.

5.      L152-153: The authors need to introduce more regarding their SVI score, as it is unclear how readers may access an English version of IMM 2018 and Mentese et al. 2019 is just a conference abstract and presentation instead of a peer-reviewed journal publication or technical report. The authors need to transparently and concisely demonstrate why their SVI can effectively measure or indicate social vulnerability of a household in the event of an earthquake. Is their SVI related to an expected loss or loss ratio given a metric of earthquake hazard strength?

**Authors' response:** Unfortunately, the previous study explaining the calculation of SoVI is available only in Turkish as an institutional report by Istanbul Metropolitan Municipality (IMM, 2018) and it was presented in a conference (Mentese et al., 2019). In the revised version, we have included a new section (*section 3.2*) to explain our data source and how SoVI was constructed in the previous study in 2017.. We have also translated IMM's work to English and included a technical summary of IMM's work as a *supplementary file 1*. We are aware that, in the literature there is a great variety of indicators that are being used to assess social vulnerability, as social vulnerability itself has a complex nature. In IMM's study, SoVI was calculated following the factor analysis framework and 53 indicators, which were reduced to 7 factors. The strategy which was proposed by Cutter et al. (2003) was used. Thus, in the construction of SoVI, social vulnerability was considered as a capacity of society and individuals to cope with hazard and damage. The indicators chosen for the calculation of SoVI have been selected following extensive literature reviews and discussions with experts. While constructing SoVI, the residential units' type and construction, and infrastructure were also used as they are potentially important in understanding social vulnerability, as they may relate

with potential economic losses, injuries, and fatalities from natural hazards. Additionally, SoVI calculated by the previous study by IMM is taken as the outcome of ML models by the present study and to verify the results or validity of it is not among the aims of the present study as made clear in *section 1, paragraph 3* and *section 3, lines 193-198.*

6.   L266-268: According to the title of the manuscript, the authors' main work was to use ML algorithms to identify predictors of social vulnerability. First, the initial feature selection of input variables of ML models has nothing to do with ML algorithms, as the authors claim clearly on L163 that the "predictors chosen have been selected following extensive literature reviews" and "discussions with experts". Then, it is still unclear what ML algorithms the authors have adopted for quantifying the importance of input variables in their predictive classification models. It seems that the main work of the authors was merely to calibrate some supervised ML classification models to map a set of already chosen input variables to their binary output variable of SVI score category. The authors need to at least explain more in a concise manner how they measured the importance of input variables of the ML models.

**Authors' response:** The statements of "predictors chosen have been selected following extensive literature reviews, discussions with experts..." relates to the selection of variables for the previously conducted survey research by IMM and SoVI calculation, as explained above in item 5. We thank the reviewer for his suggestion, this phrase does not relate to the selection of input variables for ML models and it was in the wrong place in the previous manuscript. In the revised manuscript we made it clear in *section 3.2.2.* Before fitting the ML models, we used feature selection by identifying the predictors with near zero variance and also assessed whether there exist multicollinearity or linear dependency between input variables. We also revised the selection of predictors section (*section 3.4*) accordingly.

7.  Regarding the ML classification models, I am not convinced that the authors have the capability to properly compare the prediction results of the models that they have adopted. When dealing with statistical analysis, model validation, resampling, subsampling, etc., it seems that the authors have a lot to say. But when it comes to the ML models, there is almost nothing in the manuscript. For example, what is an SVM? What is an ANN? Are the authors using the multilayer perceptron, convolutional neural network, recurrent neural network, autoencoder network, or something else for their ANN modeling? What is the difference between a CART and an RF? Are the authors capable of explaining all the models they used in their study?

**Authors' Response:** We have now included the supplementary file 3, that we did not include in the initial submission. We intended not to get into details of explaining the ML methods and packages that we have used in the main body of the manuscript, as they are commonly used approaches for binary classification problems in the ML literature. However, we explained the methodological details in the supplementary file. For ANN we have used multilayer perceptron choice, as we also made clear in the supplementary file.

8.  The entire Introduction section needs to be thoroughly revised. The authors need to make sure that their introduction is concise, relevant to their research work, and following a story line that is logically sound. For example, on L34-35, the authors start their manuscript with a UN-qualified definition of disaster in terms of coping capacity. However, this definition is irrelevant to the vulnerability quantification at a household level.

**Authors' Response:** After thorough consideration of the reviewer's comments, we have completely rewritten the Introduction and added a new section (Section 2 – Background for social vulnerability assessment).

9.      L3-37: The statement that the "evolution of an earthquake event into a disaster is typically studied through the lenses of geoscientists, civil engineers and earthquake engineers" is not true. There are many social scientists who have dedicated their research works to studying earthquake risks and disaster vulnerability to earthquakes (see, e.g., Stallings 1995 https://www.routledge.com/Promoting-Risk-Constructing-the-Earthquake-Threat/Stallings/p/book/9780202305455; Bolin and Stanford 1998 https://doi.org/10.4324/9780203028070).

**Authors' Response:** We have revised that sentence and Sections 1 and 2 to provide more details on the research works by social scientists on hazard related social vulnerability.

10.     L37-39: The statement that "it is often forgotten or ignored that the human consequences of disasters are in part derived from the composition of the population and society prior to the event" is false. There are plenty of works looking at the social factors of disaster vulnerability even for quantitative and engineering modeling purposes within the context of earthquake hazard (see, e.g., Peduzzi et al. 2009 https://doi.org/10.5194/nhess-9-1149-2009; Lin et al. 2015 https://doi.org/10.5194/nhess-15-2173-2015; Wang et al. 2019, 2020, 2021; Chen and Zhang 2022 https://doi.org/10.1016/j.ress.2022.108645).

**Authors' Response:** We have removed this section and revised the Introduction section.

11.     L49-65: This paragraph is totally unacceptable. Many sentences in this paragraph do not follow a logical flow. They read more like an awkward assemble of incompatible spare parts with fake "made in" labels on them. For example, on L55-58, the capacity of an entity to anticipate, cope with, resist, and recover from the impact of an earthquake actually includes the ability to reduce casualties due to collapse of buildings in an earthquake event.

**Authors' Response:**. Our intention in the paper is not loss assessment (also covering loss of some social elements) of disasters but we rather concentrate on the characteristics of households that are easily accessible by policy-makers which also has a place in academic literature for social vulnerability. In the revised manuscript, we explained the approaches to quantify SV in *section 2*. We again note that, while constructing SoVI, the residential units' type and construction, and infrastructure were also used by IMM as they may relate with potential economic losses, injuries, and fatalities from natural hazards.

12.     L59-61: Following the previous comment, I find it extremely difficult to understand why the authors have to talk about something called "social risks"? Also, I highly doubt that Prof. Susan Cutter has ever mentioned the term "social risks" in her 1996 paper. Can the authors provide the page number for where Cutter mentioned "social risk"?

**Authors' Response**: There is typo in that sentence referenced to Cutter (1996) which has to be social vulnerability not social risk. The social risk terminology issue and the sentence is fixed in the revised manuscript, as we have rewritten the whole Introduction

13. L66-83: This paragraph also reads awkward. It is unclear what the main point is for the authors to compile such a paragraph. On L68, the authors even cited the wrong Cutter 1996 paper.

**Authors' Response:** This paragraph is updated and we have rewritten the Introduction in the revised manuscript.

14. L86: The authors list logistic regression (LR) as a traditional data analysis tool. How could the authors justify their using LR as an ML method later?

**Authors' Response:** We thank the reviewer for his careful assessment. Logistic regression is a statistical technique which is used for binary classification problems. Due to large sample size, in our study we used a variety of supervised ML techniques for binary classification in addition to the logistic regression technique. Therefore, we had a chance to compare ML techniques to widely used logistic regression analysis. In the revised the manuscript we clarified this statement in *Section 3.5, lines 294-298.*

15. L92: The statement involving using "ML methodology over regression techniques" is confusing. Supervised ML methodology consists of two basic groups of methods. One is classification and the other is regression. ML regression methodology is part of ML methodology.

**Authors' Response:** We revised this sentence as "A relatively small number of researchers have opted to use ML methodology over traditional statistical techniques in vulnerability research". Please see *section 2, lines 166-167.*

16. L95-111: This paragraph needs to be rewritten to be concise and professional. It needs to serve the purpose of pointing out the motivation of and rationale for the proposed research. The authors need to read more technical papers published in hazard and disaster journals to get more familiar with the flavor of the introduction sections of papers that can be accepted for publication in this journal and rewrite their introduction accordingly.

**Authors' Response:** After thoroughly considering both reviewers' and editor's comments, the Introduction section have been rewritten in the revised manuscript to give concise information about the significance of the topic and contribution of the study. Then we have included Section 2, which focused on social vulnerability literature.

17. I am not at all convinced how the authors could justify the identification of risk of job loss in the event of an earthquake as a vulnerability factor/predictor. To reduce disaster risk is to reduce the expectation of event losses, which include the loss of livelihoods, or job loss. It is totally pointless to tell practitioners that, to reduce disaster risk including risk of job loss given an earthquake event, we need to reduce the risk of job loss given an earthquake event.

**Authors' Response:** In the earlier manuscript we accept that we were over optimistic to tell that our model helps to reduce disaster risk. In the revised version, we made it clear that our aim is not to predict event losses but we try to aid in assessing "who is socially vulnerable following a hazard?". Therefore, our proposed model is not directly focused on reducing disaster risk, but it would help policy makers to facilitate risk reduction activities targeted for vulnerable groups. By job loss, we meant to refer the job insecurity. Various authors discussed that, increase in the number of unemployed in a community will lower the coping capacities

and contribute to a slower recovery from the disaster (Cutter et al., 2003, https://doi.org/10.1111/1540-6237.8402002; Chen et a., 2013, https://doi.org/10.1007/s13753-013-0018-6; Llorente-Marrón et al., 2020, https://doi.org/10.3390/su12093574). We changed the terminology to "Job Insecurity" and we have described that it was measured in relation to the individuals who get involved in unregistered informal work or who are unemployed. Please see Table 2 and *section 5.3, lines 586-612* for more details.

18.     L116-118: What is this "broad conceptual model"? What are the other models that the authors have compared their model to for demonstrating "a better understanding"? How is the authors' model better?

**Authors' Response:** We revised this sentence as: "We posit that, when large data sets are available at the household level, the models developed based on ML algorithms have the potential to predict socially vulnerable households with high accuracy". Please see *section 3, lines 189-191*.

19.     L157-160: The listed three reasons for treating social vulnerability as a binary output of ML models are not convincing at all. First, with regression approaches with a numerical output variable, one can also identify vulnerability factors/predictors quantitatively and empirically. Second, the accuracy of predictions does not depend on whether using a classification or regression method. Third, it may be actually easier to interpret the regression results, especially when the regression models are linear or close to being linear.

**Authors' response:** We have included a detailed description on why we use dichotomized SoVI as an outcome of ML models. Please see *section 3.3* for our motivation to use binary SoVI and examples of other studies in the literature which used binary SVI.

20.     L497-498: Without historical data on event losses and recovery processes involved in their modeling efforts, how can the authors make such a bold statement that, based on their research, they "have found that socially, economically, and environmentally vulnerable communities are more likely to suffer disproportionately from disasters"? Where are the actual evidences?

**Authors' response:** We agree with the reviewer that we made a strong statement. And in fact, this statement is not our finding but an accepted fact in the vulnerability literature, as also pointed out by Reviewer 2. In the revised manuscript we have changed the emphasis in the sentence and added the relevant literature. Please see *Section 6, lines 627-628*.

21.     L521-526: The authors claim that their research can support decision makers and local authorities to improve disaster risk reduction practices. However, I feel hardly confident to agree with this claim after reading the manuscript. ML methods are good at predicting output variable values based on the optimization of parameters of a mathematical model that empirically represents the relationship between the input and output variables based on the data for training. What the authors have achieved is using an index-based approach to create an SVI to indicate social vulnerability during their first phase. However, as I have mentioned previously, this indicator is not social vulnerability itself. It is an indicator of social vulnerability. Without consideration of empirical data on event losses, etc., this indicator itself is not a good indicator of disaster vulnerability. Then, in their second phase, which is what is mainly presented in the manuscript, the authors used ML methods to establish models of the relationships between a set of social variables as the input and their SVI as the output. With

these models, the authors suggest that practitioners may identify pertinent social variables to improve disaster management. However, when targeting the identified social variables and changing their values, such an alteration of input variable values will only change the predicted model output value, while the changing of the model output value may have nothing to do with the actual reduction of social vulnerability. ML methodology does not identify causal relationships. I am simply wondering how, from the authors' perspective, their modeling results can actually benefit local management of seismic disaster risks. Can the authors explain it more in detail? In addition, what potential issues should the practitioners pay attention to when the practitioners are encouraged to apply the authors' models for guiding earthquake disaster risk reduction practices?

**Authors' Response**:

To start with, we want to clarify that, the work which calculates SoVI for the households was conducted by Istanbul Metropolitan Municipality and it was not the authors' own work. However, we had to rely on SoVI which was calculated by IMM as it was the only available household level SoVI available in Istanbul. We made clear that, this pre-calculated SoVI was grounded on the theoretical background which was proposed by Cutter et al. (2003, https://doi.org/10.1111/1540-6237.8402002). However, we do not intend to discuss the SoVI scores or the methodology of this previous study.

In the revised manuscript we explained the construction of SoVI in more detailed in *section 3.2.2 and supplementary file 1*. Our motivation was to assess the variables which contribute the most to the social vulnerability of the households, given a pre-developed SoVI as an outcome. We agree with the reviewer in the sense that our results are not directed towards defining disaster risk reduction policies together with a physical / social loss estimation but it is aimed at identifying the characteristics of the households that are more vulnerable to disasters using variables which are readily available in population wide data bases of various institutions. Therefore, the groups with the certain characteristics which are more vulnerable may be prioritized by decision makers in terms of their needs in order to develop new social assistance schemes that are specifically targeted to disaster vulnerability. Such kind of targeted assistance is missing in the local and national disaster risk reduction policies in Turkey, though it is a part of the Sendai Framework.

We would also like to emphasize that in a case where empirical data is not in place, we find it reasonable to use an index-based methodology based on institutionally provided databases to represent SV. We are aware that empirical findings are better in representing, but we used the index results as the empirical findings are not available for our case in Istanbul. We also think that our approach to use index-based results can be an alternative for countries where the data sources are limited. We hope that we have now clarified the misunderstandings in the revised manuscript.

**Technical Issues**

22. L24: Why are the words "Artificial", "Neural", and "Network" with their first letters capitalized while the ones on L21 are not? Plus, "(ANN)" should be following the "artificial neural network" on L21.

**Authors' Response**: We corrected the typo in the revised manuscript.

23. L42-44: Why do the authors have to mention three return periods when the mentioning of 100-year return period alone would suffice in this sentence? Also, where is the evidence to support this statement?

**Authors' Response**: We provided the literature evidence (Parsons, 2004; Utsu, 2002) and revised the sentence in the revised manuscript. Please see *section 2, lines 150-151*.

24. L53-54: What do the authors mean by the phrase "robust and concrete disaster risk reduction"? What does "robust" mean? What does "concrete" mean?

**Authors' Response:** In this sentence we intended to emphasize the importance of considering social aspects as well as physical ones when making robust and concrete recommendations, not "making robust and concrete DRR". We have corrected this typo.

25. Table 1: What is "Dept"?

**Authors' Response:** It was a typo. We have corrected this typo as "Debt".

**Responses to Reviewer 2's comments**

**Authors' Response to Overall Comments:** Firstly, we thank Dr. West for her valuable and constructive comments that improved our manuscript. We revised the manuscript, specifically we have now rewritten Introduction section and added a new section to give more details on social vulnerability literature. In addition, we have added supplementary materials which give more details about the previous work on SoVI and data sources. Please see below for a point-by-point response to reviewer's comments and concerns.

**Specific Suggestions:**

1. An overarching question I would like the authors to address is whether, and how, the ML analyses might be used to improve upon the original SVI measure for Istanbul. What do the ML models add to the understanding of social vulnerability in Istanbul that was not clear previously? I think the manuscript will be strengthened if the authors can better connect these dots for readers.

**Authors' Response:** We thank the reviewer for this comment, giving us the chance of elaborating on this topic. In this study, we used the pre-constructed SoVI (based on the survey on 41,093 households) as the output, which the reviewer refers to as the pre-constructed SoVI. In that previous study (conducted by Istanbul Metropolitan Municipality), the SoVI score and SV level of the households were calculated based on pre-determined variables identified through a literature review. These variables were related to social, economic and demographic properties of the households as well as cultural characteristics of the community, such as the perception of and preparedness against earthquake risk based on Cutter et al.'s (2003, https://doi.org/10.1111/1540-6237.8402002) framework. The required information to represent these predictors were obtained via survey research. For more details, please see *sections 3.2.2, and the supplementary file 1*. In this paper, we aimed to propose a modelling strategy that can predict the SV level of these households using easily accessible data without requiring to implementation of a household-based survey. For this modelling approach, we used pre-constructed SoVI of the households as an outcome of the models. As we make use of easily accessible large household data, we aid in reducing the time and economic burden that one may spend conducting surveys to calculate household level SoVI and assess households with high SV. Our modelling approach could also be used by decision makers to identifying and prioritizing action towards target groups in the population in the interests of risk mitigation, by

classifying the household with respect to their SV level. In this regard, we believe such an ML approach to identify SV is beneficial to effectively and practically interpret the social context better in the face of disasters.

We also believe that such practicality will make it possible to understand SV better as it will enable researchers to adopt this study on different spatial contexts with different variables. Eventually, this will lead to a more comprehensive understanding of the phenomenon. We believe that we clarified our aim and scope in the revised manuscript in *Section 1- Introduction, paragraphs 3 and 4*, which was not clear in the previous version.

2. I recommend building upon the discussion of the *scale* at which data were analyzed as a strength of this research. Social vulnerability is not often able to be analyzed at the household level, so that is a significant potential contribution of this research worth discussing.

**Author's response:** We appreciate the reviewer for drawing attention to the importance of measuring social vulnerability at the household level, which lacks in the literature due to limited data availability. In that sense, our study is one of the few studies which is based on large-scale household survey to assess social vulnerability predictors, particularly in developing countries (Debesai. 2020, https://doi.org/10.1088/1757-899x/1001/1/012093). In the revised manuscript, we emphasized this major contribution of our study by discussing the importance of assessing household level SV in Section 1 and provided details on the methodology of the survey (which was our data source) and its outputs in *Section 3.2*.

3. The current description of the data and methods used to construct the Istanbul SVI are not yet sufficiently complete and accurate to allow their reproduction by fellow scientists. As I understand, there is a vulnerability index for Istanbul that was created as a Phase 1 of this study. However, its description is not available in English (the language of this journal), and there seems to be no peer reviewed publication associated with the SVI. In light of this, the authors need to describe the construction of the index in detail before using it as the outcome variable in the ML models. Thus, I recommend adding a section to the manuscript describing the construction of the social vulnerability index. I encourage the authors to acknowledge the limitations of various approaches to index construction, including those raised by Spielman et al. (2020) and others. I include some recommended publications on this topic at the end.

**Author's response:** Unfortunately, the previous study explaining the calculation of SoVI is available only in Turkish as an institutional report by Istanbul Metropolitan Municipality (IMM, 2018) and it was presented in a conference (Mentese et al., 2019). In the revised version, we have included a new section (*section 3.2.2*) to explain how SoVI was constructed in the previous study. We have also translated IMM's work to English and included a detailed summary of IMM's work as a *supplementary file 1*. We are aware that, in the literature there is a great variety of indicators that are being used to assess social vulnerability, as social vulnerability itself has a complex nature. In IMM's study, SoVI was calculated following the Cutter et al.'s (2003) factor analysis framework and 53 indicators, which were reduced to 7 factors. Thus, in the construction of SoVI, social vulnerability was considered as a capacity of society and individuals to cope with hazard and damage. The indicators chosen for the calculation of SVI have been selected following extensive literature reviews and discussions with experts. Additionally, SoVI calculated by the previous study by IMM is taken as the outcome of ML models by the present study and to verify the results or validity of it is not among the aims of the present study as made clear in *section 3, paragraph 2*. In the revised manuscript, we have added a new section (*section 2*) that summarizes the approaches for calculating SVI, including the discussion of various approaches in the literature. As our aim was not to construct SVI in this study -as we have used pre-constructed SoVI – we intended to keep this discussion short. We note that, we are grateful to the reviewer for the publication suggestions on SVI and we have benefit from those publications to a great extend while writing up this discussion.

4. Building upon the explanation of the SVI construction, please also explain why the decision was made to evaluate the vulnerability index as a binary variable that refers to the top 20% of the vulnerability index. For instance, why not use the continuous index as the outcome variable? Why not use the top 25%? Finally, discuss any potential strengths, weaknesses, or consequences of defining vulnerability in this way.

**Authors' response:** In the previous work, SoVI's of the households were classified into four categories as explained in the supplementary file 1: Low vulnerability ($< -1$ SD), Low-moderate Vulnerability (-1 to 0 SD), High-moderate vulnerability (0 to 1 SD), and High Vulnerability ($> 1$ SD). In the dataset that was used in here, instead of using four SVI categories as an outcome, we used binary SVI (High Vulnerability ($> 1$ SD) vs all others) to discriminate the households that requires the most urgent action from all others, and this group corresponds to approximately 20% of the households. As suggested by Cutter and Finch (2008,

https://doi.org/10.1073/pnas.071037510), SoVI score does not have any unit and rather than its absolute value, its importance lies within its comparative value. Therefore, we relied on this definition in accordance with our aim. Also statistically speaking, the available performance metrics for a multi-class confusion matrix are limited compared to a binary classification problem (Markoulidakis et al., 2021, https://doi.org/ 10.3390/technologies9040081). Therefore, in addition to our motivation of identifying households with high SV, for the sake of interpretability and ease of application we used binary outcome. Please see *section 3.3* for more details and examples of various studies that use binary SoVI.

5.  It was also my impression that the variables in the Istanbul vulnerability index are not specific to *earthquake* vulnerability in particular. Instead, they seem to refer to social vulnerability more generally and not in relation to any single hazard. This is not necessarily a problem with the data. However, the claim to understand earthquake vulnerability in particular needs to be more fully substantiated, or removed, because the SVI data used in this study do not appear to refer to earthquake-related vulnerability, even if the original household survey did focus on earthquakes.

**Authors' response:** The analysis presented in this research is based on a survey data, which was carried out by the "the Directorate of Earthquake and Ground Research of Istanbul Metropolitan Municipality". The data available were collected through face-to-face interviews conducted with the selected households. The interview questions used to derive SoVI include questions related to respondents', housing conditions, preparedness to earthquake, their risk of perception to earthquakes and their past experience on earthquake incidence (please see *supplementary file 1*). We agree with the reviewer in the sense that, although the SoVI was focused on earthquakes in the previous study, the indicators that we used for predicting SV in the present study can be implemented  more generally to assess the possible impact of social vulnerability due to other types of hazards. Therefore, we do not intend to claim that selected variables are specific to earthquake vulnerability, and our model can be applied to any major hazard that can affect macro (i.e. roads, bridges, public buildings, etc.) and micro (i.e. housing) infrastructure. We made this clear in *Section 3.4, lines 263-265*.

6.  I am curious to know how the "risk of job loss" was assessed, and I would be interested to see more explanation of why job loss would be specific to a post-earthquake context, or whether this is a general measure of job insecurity. If possible, please describe briefly

how that question was asked on the original survey and whether it was a self-assessment of potential job loss. This will hopefully help readers better understand that variable.

**Authors' response:** We thank reviewer for her suggestion which have helped us to clarify any misunderstanding regarding job loss variable. By job loss, we meant to refer the job insecurity. Various authors discussed that, increase in the number of unemployed in a community will lower the coping capacities and contribute to a slower recovery from the disaster (Cutter et al., 2003, https://doi.org/10.1111/1540-6237.8402002; Chen et a., 2013, https://doi.org/10.1007/s13753-013-0018-6; Llorente-Marrón et al., 2020, https://doi.org/10.3390/su12093574). We changed the terminology to "Job Insecurity" and we have described that it was measured in relation to the individuals who get involved in unregistered informal work or who are unemployed. Please see Table 2 and *section 5.3, lines 586-613* for more details.

7. The last sentence of the abstract suggests that "The machine learning methodology and the findings that we present in this paper can serve as a guidance for decision makers…" I would like to know more about how specifically the machine learning methodology could be used by decision makers. I do understand how the SVI data can be a tool for decision makers, but it is not yet clear to me how the ML component could practically be used by decision makers to reduce vulnerability. Would you argue that this ML analysis can be used to improve the SVI as a decision making tool? If so, explaining how and providing an example of a use case might be helpful.

**Authors' response:** We thank the reviewer for giving us a chance to clarify this important aspect of our study. Our motivation was to assess the variables which contribute most to the social vulnerability of the households, given a pre-developed SoVI as an outcome. By using our proposed ML model, one can input these variables – which are easily accessible by policy-makers – to our proposed final ANN model and obtain the status of social vulnerability of the household. Hence our model proposal has potential to reduce the time and economic burden that one may spend to conduct surveys to calculate household level SVI and assess households with high SV. We agree in the sense that our results are not directed towards defining disaster risk reduction policies but based on the social vulnerability characterization, these households can be prioritized by decision makers to develop new social policies targeted to disaster vulnerability. Such targeted policies are particularly missing in the Turkish context as well as in many of the low-mid income countries. We believe that, we clarified the contribution of

using ML methodology in relation to predict SV in the revised manuscript. Please see *section 1* for the scope of the study and *section 7* for more details regarding how our model can be used in practice and who are the stake holders.

**Grammatical/proofreading:**

1. In the abstract, please define what the outcome variable is when mentioning it.

**Authors' response:** We have defined the outcome in the abstract in the revised manuscript.

2. Change "dept" to "debt" in the last line of Table 1

**Authors' response:** It was a typo. We corrected this typo as "Debt".

3. I recommend avoiding the term "natural disasters" and instead simply using "disasters" or being more specific. (line 63, 103, 453, 529)

**Authors' response:** We used "disasters" in the revised manuscript instead of "natural disasters".

4. Page 4, Line 121: Avoid using the word "intrinsic" to describe social vulnerability because social science research specifies that vulnerability is *not* intrinsic to people; it is instead a condition borne by some people under certain conditions. People are also not vulnerable at all times or in all contexts. In other words, social vulnerability does not emerge from the characteristics in the SVI. Rather these variables can sometimes indicate or signal who is more likely to bear vulnerability created by structural forces and power imbalances. It is important that the language used is clear about this.

**Authors' response:** We totally agree with the reviewer and we removed the word "intrinsic" in the revised manuscript.

5. Page 25, Line 497: I recommend rephrasing the sentence, "We have found that socially, economically, and environmentally vulnerable communities are more likely to suffer disproportionately from disasters," because this is not a finding of the current study as it does not evaluate impacts of disasters. Instead, this is a finding of many previous

studies that you could perhaps cite here. Simply removing the words "we have found that…" may be sufficient.

**Authors' response:** We agree that we made a strong statement here. We revised this sentence by removing "we have found that…" and added references to support this sentence. Please see *Section 6, Lines 626-627.*

6. I find the phrase "social vulnerability risk" to be a bit confusing and inconsistent with most literature on this topic. Social vulnerability is typically considered a sub-component of risk, so these phrases should not be combined. Instead, use the phrase "social vulnerability" without the word "risk." For similar reasons, the phrase "social risk" should be replaced with either "social vulnerability" or "disaster risk," as appropriate.

**Authors' response:** We thank the reviewer for this suggestion. We agree that "social vulnerability risk" was not compatible with the literature and we used phrase "social vulnerability" in the revised version. Also, there was typo in the sentence related to "social risk" referenced to Cutter (1996). It has to be "social vulnerability" as the reviewer has suggested, not "social risk". The social risk terminology issue and the sentence is fixed in the revised manuscript.

.

---

## Referee Report (RR1)

**Review on nhess-2022-198-v2: Using machine learning algorithms to identify predictors of social vulnerability in the event of a hazard: Istanbul case study**

After reviewing the materials provided by the authors for this submission, I applaud the authors' efforts in revising their manuscript and offering exciting responses to the previous comments of the reviewers. However, at this stage, I still do not feel that the current version of the manuscript is good enough for publication in NHESS. I still have a number of questions and concerns at the medium or minor level. The authors need to address them. In addition, I strongly suggest the authors, when finishing this round of revision, carefully read their modified manuscript and make serious efforts in polishing their writing and patiently conducting some editing works to their manuscript. Below are listed my medium and minor questions and concerns.

**Medium and Minor Issues**

1. L23: "CART" is short for "classification and regression tree". Please use either "classification tree (CT)" or "classification and regression tree (CART)" to avoid confusion. The same for the rest of the manuscript.

2. L40: When used as a countable noun, "vulnerability" usually means a weak link, a loophole, a fragile element, etc., of a system that may be exploited by hazardous agents to result in loss to the system. Is that what the authors mean here? If not, please use the uncountable version of the noun "vulnerability".

3. L86: I suggest removing "in fact" because the statement here is still about an intellectual guess or belief.

4. L150-151: Very large earthquakes (over Mw7.0) do seem to be rare for Istanbul. However, earthquakes with a magnitude 4 or above can still cause significant damage to communities (see, e.g., Wang and Sebastian 2022 https://doi.org/10.5194/nhess-22-4103-2022). These earthquakes shouldn't be rare at all according to the estimated 100-year return period for an earthquake with Mw7.0 and above around Istanbul. In addition, as the manuscript has changed its focus from earthquake to all hazards, the large hazardous events should be more frequent than merely large earthquakes. Even the authors themselves mention on L203-204 that the study area "is in a region that is prone to natural hazards where a large-scale disaster happens every seven to eight years (Baris, 2009)". Moreover, at the household level, many families do not have to wait for a large-scale disaster to occur before experiencing loss unfortunately. More impacts to households are likely to be caused by the much more frequent smaller-scale hazardous events that may not even be considered or defined as disasters.

5. L202-210: This paragraph is inappropriate for hazards in general. Please revise it.

6. L213-214: Since the focus is on hazards in general instead of earthquake now, I suggest the authors explain a little bit regarding why the survey conducted by an earthquake-related organization can be used for all hazards.

7. L231-234: There are grammatical problems associated with this sentence "It considers … and socio-economic status". Please modify it.

8. L303-305: The authors claim that for "different tuning parameter alternatives, the choice of the optimal tuning parameter was determined by the largest area under the curve (AUC) value of the receiver operating characteristic (ROC) curve using the automated grid search". However, in the supplementary file 3 p. 2, the authors clearly state that the parameter K of KNN is "determined with the square root of the number of points in the training data set". These two statements are inconsistent with each other. Why? Moreover, the parameter **ntree** of RF is also determined arbitrarily by the authors without grid search.

9. L403-404: The sentence "The prevalence … among 41,093 households" needs to be modified for all hazards.

10. L422-424: Sensitivity and recall are the same thing. Positive prediction value and precision are the same thing.

11. L435: Fig. 3 may not be friendly enough to colorblind readers.

12. L468-472: It is still unclear in the manuscript how the relative importance of predictors is measured. What methods or algorithms do the authors use for quantifying predictor importance?

13. L475: In Fig. 4, it is unclear whether it is the size of the circle or the hue that is supposed to correspond to the number next to the circle in the legend. Also, the scale of the circle size does not cover the small value as for debt in Fig. 4A. In addition, what do the colors of the bars in Fig. 4b mean? If they indicate the variable importance in terms of percentages, then these colors provide redundant information that is confusing.

14. L506-507: I find it difficult to see the connections between this sentence "For many decades…derived variables (Di Franco and Santurro, 2020)" and the rest part of this paragraph. I suggest the authors make modifications to the paragraph.

15. L513-521: This paragraph reads awkward. What the main point is here is unclear. The authors need to revise the paragraph. Also, it is dangerous to assume that the trained non-linear structure of the neurons of an ANN represents well the relationships between the input variables. Every training may result in a totally different internal structure of the ANN.

16. L530-531: Why is it important whether the training data has to be balanced? If the identification of high social vulnerability is preferred, why don't the authors use a reversely imbalanced training data to boost the sensitivity, etc., even more?

17. L538-549: What is the main theme of this paragraph? Are the authors trying to discuss the methods for measuring importance of input variables or the important input variables identified in their study? Also, as I have asked previously, what is actually the method or algorithm that the authors use for their quantification of variable importance?

18. L538-585: The writing of these 3 paragraphs needs to be improved.

19. L566-568: It does not make sense that prevalence of social security in low vulnerability households is lower than in high vulnerability households.

20. L569-585: The material in this paragraph involving earthquake needs to be modified to fit the tone for all hazards.

21. L624: What are "hazard risks"? I suggest the authors use "hazards" here instead.

22. L641-642: Why "fault lines"? Is the manuscript supposed to be for all hazards now?

23. The authors' final ANN model has an accuracy of less than 75%. That is quite low for identifying high vulnerability households. Isn't this an obvious limitation? If I were the public official to look at vulnerable households, I would definitely need a prediction model with accuracy over 90%, or even 95% or 99%. How could I afford to miss those actually vulnerable households while providing a lot of resources to households that are actually not highly vulnerable?

24. Supplementary file 3, p.2, 4. SVM: Why do the authors use the radial basis function (RBF) kernel? A linear kernel can correspond better to a hyperplane in a vector space with the same number of dimensions as the number of input variables. When an RBF kernel is applied, it is equivalent to transforming the original vector space into a vector space with an infinite number of dimensions. Such a transformation can be demonstrated with the application of a Taylor expansion when we are using the Lagrange multipliers to solve the optimization problem for calibrating the SVM.

25. Supplementary file 3, p.2, 7. ANN: ANN is not necessarily non-linear. When all the activation functions are linear, an ANN is equivalent to a linear model. ANN is also not necessarily based on deep learning. Deep learning involves an ANN with at least 2 hidden layers. If the authors only use 1 hidden layer, that is not deep learning. Also, what activation functions do the authors use for their multilayer perceptron ANN model?

---

## Editor Decision (ED1)

Dear Dr. Kalaycioglu and co-authors

Thank for submitting your manuscript to our Special Issue on the use of Machine Learning in Natural Hazards Risk Assessment. We have received comments from two reviewers who are intrigued by the idea of applying machine learning to identify predictors of social vulnerability but would like to see major revisions made to ensure the manuscript can be accepted for publication. To be considered for future publication in this SI, the following points from the reviewers must be addressed.

- **Providing an improved description, literature review, and discussion on social vulnerability indices, methods, and theory.** Both reviewers pointed out several areas of research that are not acknowledged in the manuscript, including empirical modeling for social vulnerability, critiques of social vulnerability indices, the underlying theory for social vulnerability, and social science research in the event of an earthquake, among many others. The authors should engage with this literature throughout the manuscript— in the introduction, methods, results, and discussion. This also includes using the correct terminology for social vulnerability research and avoiding phrasing that contradicts basic concepts of social vulnerability.
- **Addressing comments on the proposed framework to predict SVI (an index) with household-level social variables.** Both reviewers would like to read justification of this approach. This broadly requires a thorough description of using SVI as the predicted outcome (as opposed to other measures), and ensuring that SVI is not equated with disaster impact/losses/recovery. The selection of predictor variables should be described, including more detail on why/how they were selected, how they were developed, and whether they'd be available at the household level in order to apply this approach in the future. A practical comparison between this approach and the traditional SVI approach in Istanbul should also be completed. Methodologically, this includes a detailed description of the construction of the SVI score that was used as an outcome since the documentation is not easily accessible and addressing concerns about using this as a binary outcome.
- **Addressing concerns about whether this approach is specific to earthquakes.**
- **Clarifying, in a thorough manner, how this approach could be implemented by stakeholders.** This includes clearly stating who those stakeholders are, how they would apply the ML approach proposed in this manuscript, and a potential use case.

**Sabine Loos, PhD**
Editor | NHESS Special Issue on Advances in machine learning for natural hazards risk assessment

---

## Author Response (AR2)

Dear Editor Dr. Sabine Loos,

We would like to thank you and Reviewer 1 for re-evaluating our revised manuscript. We appreciate the time and effort that you have dedicated to providing feedback. We hope that our revisions will satisfy the proposed requirements for successful publication. Please see below for a point-by-point response to your and Reviewer 1's comments and concerns.

Yours sincerely,
Oya Kalaycioglu, PhD, on behalf of all authors

**EDITOR'S COMMENTS**

1. Explaining methods for quantifying variable importance – Because the motivation of this paper is to understand which factors influence social vulnerability by using Machine Learning models, it is important to more thoroughly understand how variable importance is quantified for all models used beyond the references included in the text. Please add more description in the Methods section.

**Authors' response:** Thank you for this valuable suggestion which helped us to improve the methodology section of our manuscript. We have included a new sub-section "*3.8 Variable importance analysis*" which provides details about how the ML algorithms and logistic regression determine the importance of variables.

2. Earthquake-specific versus all-hazards vulnerability – Additional justification is required that this model is relevant for all-hazards rather than specific to earthquakes, especially since the dataset that was used is an earthquake-specific household survey. Corresponding revisions are necessary in the manuscript.

**Authors' response:** Thank you for this comment. In the revised version, we have made the relevant revisions to make clear that our model is applicable for hazard related vulnerability. Cutter et al. (2003, Social Vulnerability to Climate Variability Hazards: A Review of the Literature, p.23) reported that "*... the accepted theoretical understanding that social vulnerability is independent of hazard type. Zones of differential exposure to any or all hazards combine with SoVI to create place vulnerability (for example see Borden et al. 2007; Burton and Cutter 2008; Wood et al. 2009)"*. Please see relevant information in subsection *3.2.2 Construction of SoVI*, p.9. L239-242. Therefore, we assumed that earthquake related data collected in household survey and the indicators used for SoVI in Istanbul are applicable to other hazard events as well (please see L214-217 and L268-270).

3. Justification of ML models' performance – Can the authors justify the reported model performance for the ANN (and other ML models used) and why this is acceptable, perhaps by comparing to similar studies?

**Authors' response:** Following your and the Reviewer 1's comments on our model accuracy, we have revised the sub-section "*5.1 The selection of the optimal ML method*". In the revised

version, we have now included the discussion on our optimal ANN model's (and other models') performance. We compared the performance metric values we obtained with our ML models to the acceptable values in the literature. We considered our proposed ANN model to have a good discriminative ability (AUC>0.80) according to Hosmer et al.'s (2013) criteria. We also considered the accuracy of our optimal ANN model (which is 73%) to be acceptable as the value is halfway between 50%, which is useless, and 100%, which is perfect (Power et al., 2013). There is a limited number of studies in the literature that have used ML to predict hazard-related social vulnerability and reported performance metrics. We also included the performance metrics they have reported. We discussed that although our models' accuracies were relatively lower compared to other studies which assessed social vulnerability to hazards with machine learning techniques, our approach can be useful for decision-makers to take immediate action for the most vulnerable households. We also noted that, our models can further benefit by incorporating more predictor variables.

**REVIEWER 1' COMMENTS**

**Medium and Minor Issues**

1. L23: "CART" is short for "classification and regression tree". Please use either "classification tree (CT)" or "classification and regression tree (CART)" to avoid confusion. The same for the rest of the manuscript.

**Authors' response:** We revised the terminology and used "classification and regression tree (CART)" throughout the text and made the correction in L23.

2. L40: When used as a countable noun, "vulnerability" usually means a weak link, a loophole, a fragile element, etc., of a system that may be exploited by hazardous agents to result in loss to the system. Is that what the authors mean here? If not, please use the uncountable version of the noun "vulnerability".

**Authors' response:** We corrected the typo in the revised manuscript and used uncountable version of the noun vulnerability in L40.

3. L86: I suggest removing "in fact" because the statement here is still about an intellectual guess or belief.

**Authors' response:** We agree with the reviewer's suggestion and removed "in fact" in L86.

4. L150-151: Very large earthquakes (over Mw7.0) do seem to be rare for Istanbul. However, earthquakes with a magnitude 4 or above can still cause significant damage to communities (see, e.g., Wang and Sebastian 2022 https://doi.org/10.5194/nhess-22-4103-2022). These

earthquakes shouldn't be rare at all according to the estimated 100-year return period for an earthquake with Mw7.0 and above around Istanbul. In addition, as the manuscript has changed its focus from earthquake to all hazards, the large hazardous events should be more frequent than merely large earthquakes. Even the authors themselves mention on L203-204 that the study area "is in a region that is prone to natural hazards where a large-scale disaster happens every seven to eight years (Baris, 2009)". Moreover, at the household level, many families do not have to wait for a large-scale disaster to occur before experiencing loss unfortunately. More impacts to households are likely to be caused by the much more frequent smaller-scale hazardous events that may not even be considered or defined as disasters.

**Authors' response:** We thank the reviewer for this comment. We agree with the reviewer that smaller-scale hazardous events can cause losses at the household level as well. As this paragraph relates the pros and cons of using historical data to calculate SoVI, we mentioned that when catastrophic hazard occurrence is rare, the policy-makers can underestimate the impacts of a major hazard event, if they rely on historical data from the smaller-scale hazardous events where the losses are much less due to infrastructural investments. For example, in İstanbul – Türkiye, primarily an earthquake-prone zone, using empirical loss data of frequently occurring small-scale hazard events may mask the possible impacts of a major earthquake (over 7.0 Mw), which is rare due to historical records. Please see the relevant revision in L148-155.

5. L202-210: This paragraph is inappropriate for hazards in general. Please revise it.

**Authors' response:** We thank the reviewer for this suggestion. We revised sub-section *3.1 Study Area* to reflect hazards in general and provided more relevant information about Istanbul (L194-211).

6. L213-214: Since the focus is on hazards in general instead of earthquake now, I suggest the authors explain a little bit regarding why the survey conducted by an earthquake-related organization can be used for all hazards.

**Authors' response:** We made clear that the household survey data that we have relied on is collected by İstanbul Metropolitan Municipality (IMM) in 2017 to assess *disaster-related* social vulnerability of the households in İstanbul. The variables used in this research were in line with the social science and disaster literature, where such research is focused generally on the social factors that increase or decrease the impact of specific hazard events on the local population (L214-217).

7. L231-234: There are grammatical problems associated with this sentence "It considers … and socio-economic status". Please modify it.

**Authors' response:** To modify this sentence, we advised to a native English speaker. Following his suggestion, we modified this sentence. (L233-235).

8. L303-305: The authors claim that for "different tuning parameter alternatives, the choice of the optimal tuning parameter was determined by the largest area under the curve (AUC) value of the receiver operating characteristic (ROC) curve using the automated grid search". However, in the supplementary file 3 p. 2, the authors clearly state that the parameter K of KNN is "determined with the square root of the number of points in the training data set". These two statements are inconsistent with each other. Why? Moreover, the parameter **ntree** of RF is also determined arbitrarily by the authors without grid search.

**Authors' response:** In the previous version of the supplementary material, we made a typo regarding the explanation for determining the tuning parameters. As we have stated in the manuscript, we used automated grid search to find the optimal tuning parameters. Accordingly, we revised the supplementary file 3. p.1-2, and also clearly stating that we determined tuning parameters with grid search for all methods in the manuscript (L291-292).

9. L403-404: The sentence "The prevalence … among 41,093 households" needs to be modified for all hazards.

**Authors' response:** We revised the sentence accordingly (L413).

10. L422-424: Sensitivity and recall are the same thing. Positive prediction value and precision are the same thing.

**Authors' response:** We have removed the terms which imply the same metric in the manuscript (L434). However, we kept both terminologies in the R-shiny app, to help the readers who are familiar with different terminology.

11. L435: Fig. 3 may not be friendly enough to colorblind readers.

**Authors' response:** We thank the author for this suggestion. We have revised the colors in Figure 3, according to R color guidelines which provide a color palette for colorblind readers.

12. L468-472: It is still unclear in the manuscript how the relative importance of predictors is measured. What methods or algorithms do the authors use for quantifying predictor importance?

**Authors' response:** We thank the reviewer for this suggestion, which helped us to improve the methodology section of our manuscript. We have included a new sub-section "*3.8 Variable importance analysis*" (p15, L370-390), which provides details about how the ML algorithms and logistic regression determine the importance of variables.

13. L475: In Fig. 4, it is unclear whether it is the size of the circle or the hue that is supposed to correspond to the number next to the circle in the legend. Also, the scale of the circle size does not cover the small value as for debt in Fig. 4A. In addition, what do the colors of the bars in

Fig. 4b mean? If they indicate the variable importance in terms of percentages, then these colors provide redundant information that is confusing.

**Authors' response :** We thank the review for this suggestion. We corrected the circle size of the "debt" variable. Also, we want to clarify that both the size and the color of the circles in Fig. 4A represent number next to the figure legend. We also plotted this figure using either only varying the size or color of the circles. But as the circles were overlapping for some variables, it was hard to interpret the difference between the variables when only one of these strategies (i.e., size or color) was used. Therefore, for Fig. 4A we kept the version where both size and colors varied. However, for Fig 4B, we agree with the reviewer and in the revised version we used the same color for the bars. In the previous version, the gradient color was used as an indication of lower percentage, but as these percentages were already provided in the figure we agree that varying colors were confusing.

14. L506-507: I find it difficult to see the connections between this sentence "For many decades…derived variables (Di Franco and Santurro, 2020)" and the rest part of this paragraph. I suggest the authors make modifications to the paragraph.

**Authors' response**: To focus on discussing our findings from our ML models, we removed the first sentence starts with a comment on data analysis and social sciences. We revised this paragraph (L514-523)

15. L513-521: This paragraph reads awkward. What the main point is here is unclear. The authors need to revise the paragraph. Also, it is dangerous to assume that the trained non-linear structure of the neurons of an ANN represents well the relationships between the input variables. Every training may result in a totally different internal structure of the ANN.

**Authors' response**: Following the Editor's and Reviewer 1's comments, we have revised the sub-section "*5.1 The selection of the optimal ML method*". In the revised version, we have now included the discussion on our optimal ANN model's (and other models') performance. We replaced this paragraph with a new paragraph comparing our results from the ANN model with the results of different studies. For these comparisons we focused on AUC and accuracy as an indication of the model performances (L524-539).

16. L530-531: Why is it important whether the training data has to be balanced? If the identification of high social vulnerability is preferred, why don't the authors use a reversely imbalanced training data to boost the sensitivity, etc., even more?

**Authors' response:** The consequence of ignoring the issue of imbalanced data is to over-predict the class with higher frequency (Esposito et al., 2021, doi.org/10.1021/acs.jcim.1c00160 J). This increases specificity, and therefore reduces sensitivity. In our case, that results in over-predicting low vulnerability group, thus increasing specificity. However, our aim was to increase sensitivity to identify the households with high social vulnerability more accurately. The sensitivity increased when we used under sampling, which discards data points from the

majority class (i.e. low vulnerability group) at random until a more balanced distribution is reached while training models. Please see L546-552 for relevant explanation.

17. L538-549: What is the main theme of this paragraph? Are the authors trying to discuss the methods for measuring importance of input variables or the important input variables identified in their study? Also, as I have asked previously, what is actually the method or algorithm that the authors use for their quantification of variable importance?

**Authors' response:** We revised this paragraph. We know included a sub-section "*3.8 Variable importance analysis*" which provides details about how the ML algorithms and logistic regression determine the importance of variables. In this section, we know only discuss the important input variables identified with our models.

18. L538-585: The writing of these 3 paragraphs needs to be improved.

**Authors' response:** We have revised these 3 paragraphs. (L559-598).

19. L566-568: It does not make sense that prevalence of social security in low vulnerability households is lower than in high vulnerability households.

**Authors' response:** It was a typo and we corrected in the revised manuscript. (L581-582)

20. L569-585: The material in this paragraph involving earthquake needs to be modified to fit the tone for all hazards.

**Authors' response:** We have edited the paragraph to be applicable to all hazards. (L583-598)

21. L624: What are "hazard risks"? I suggest the authors use "hazards" here instead.

**Authors' response:** It was a typo and we corrected in the revised manuscript. (L633)

22. L641-642: Why "fault lines"? Is the manuscript supposed to be for all hazards now?

**Authors' response:** We removed the sentence, which emphasises the fault lines, as the manuscript is focused on all hazards now.

23. The authors' final ANN model has an accuracy of less than 75%. That is quite low for identifying high vulnerability households. Isn't this an obvious limitation? If I were the public official to look at vulnerable households, I would definitely need a prediction model with accuracy over 90%, or even 95% or 99%. How could I afford to miss those actually vulnerable households while providing a lot of resources to households that are actually not highly vulnerable?

**Authors' response:** Our final proposed ANN model had an AUC >0.80 which indicates a good level of discriminative ability between households with high and low social vulnerability according to Hosmer et al.'s (2013) criteria. We also considered the accuracy of our optimal ANN model (which is 73%) to be acceptable as the value is halfway between 50%, which is useless, and 100%, which is perfect (Power et al., 2013). However, the accuracy of our models were relatively smaller compared to the similar studies which use ML models to predict SV (Abarca-Alvarez et al., 2019, doi.org/10.3390/ijgi8120575; Alizadeh et al., 2019, doi.org/10.3390/su10103376). That was due to the fact that, we used quantifiable household data as our aim in this manuscript was to present an optimal modelling strategy capable of processing readily available large databases. Our approach can be useful for decision-makers to take immediate action for the most vulnerable households, and further can be enhanced by incorporating more predictor variables. We have revised the sub-section "*5.1 The selection of the optimal ML method*" accordingly and added the limitation of relatively low model accuracy to section *"6* limitations and recommendations" in the revised manuscript. (L654-656).

24. Supplementary file 3, p.2, 4. SVM: Why do the authors use the radial basis function (RBF) kernel? A linear kernel can correspond better to a hyperplane in a vector space with the same number of dimensions as the number of input variables. When an RBF kernel is applied, it is equivalent to transforming the original vector space into a vector space with an infinite number of dimensions. Such a transformation can be demonstrated with the application of a Taylor expansion when we are using the Lagrange multipliers to solve the optimization problem for calibrating the SVM.

**Authors' response:** For training data with SVM, we have fitted three possible versions of kernel functions which are radial kernel, polynomial kernel and linear kernel. As mentioned in supplementary file 3, p.2, 4. SVM, line 6-7, radial kernel is used in our study as it provided larger AUC compared to using linear or polynomial kernel functions. Thus, as we obtained the highest discriminative ability between the households with high and low SV when we used RBF function.

25. Supplementary file 3, p.2, 7. ANN: ANN is not necessarily non-linear. When all the activation functions are linear, an ANN is equivalent to a linear model. ANN is also not necessarily based on deep learning. Deep learning involves an ANN with at least 2 hidden layers. If the authors only use 1 hidden layer, that is not deep learning. Also, what activation functions do the authors use for their multilayer perceptron ANN model?

**Authors' response:** We thank the reviewer for this comment. We made the following revisions in the supplementary file 3, p.2, 7. ANN method:

> **7. Artificial Neural Network (ANN),** which is capable of learning any non-linear function, is  created by imitating the functioning of the human brain and transferring it to the computer environment, which is first proposed by McCulloch and Pitts (1946).  The mechanism operates with

artificial neurons that form input and output neurons and a hidden layer(s), which is frequently used for the data set that cannot be separated linearly. When there are multiple layers between the input and output layers, ANN is as a deep neural network (DNN) (Schmidhuber, 2015). Although computationally expensive, it is successful to detect complex nonlinear relationships between variables. We used sigmoid/logistic function as the activation function, which is commonly used to add non-linearity to an ANN model. Additionally,  the multilayer perceptron choice was employed for ANN model by adapting **nnet** method in **caret**, which contains two tuning parameters: number of neurons in a hidden layer and decay parameter that controls initial weights for input neurons.

---

## Editor Decision (ED2)

Dear Dr. Kalaycioglu and co-authors,

Thank you for your revisions to your manuscript and additional Supplementary Files/Apps. Based on Reviewer 1's re-review, I am recommending this manuscript for Minor Revisions.

Please see Reviewer 1's additional comments and address those. Primarily, please pay attention to the following overarching areas:

1. Explaining methods for quantifying variable importance – Because the motivation of this paper is to understand which factors influence social vulnerability by using Machine Learning models, it is important to more thoroughly understand how variable importance is quantified for all models used beyond the references included in the text. Please add more description in the Methods section.
2. Earthquake-specific versus all-hazards vulnerability – Additional justification is required that this model is relevant for all-hazards rather than specific to earthquakes, especially since the dataset that was used is an earthquake-specific household survey. Corresponding revisions are necessary in the manuscript.
3. Justification of ML models' performance – Can the authors justify the reported model performance for the ANN (and other ML models used) and why this is acceptable, perhaps by comparing to similar studies?

Thank you,

**Sabine Loos, PhD**
Editor | NHESS Special Issue on Advances in machine learning for natural hazards risk assessment

---

## Author Response (AR3)

Dear Handling Editor Dr Sabine Loos and Executive Editor Prof Philip Ward,

We would like to thank you for accepting our revised manuscript with corrections. We appreciate the time and effort that you have dedicated to providing feedback. Please see below for the corrections we did based on your comments.

Yours sincerely,
Oya Kalaycioglu, PhD, on behalf of all authors

**Corrections:**

1- We have removed the unnecessary references and text that relates to earthquake hazard in Istanbul, as the direction of the paper is hazards in general.

2- We have corrected the typos (e.g., R Shiny, subsampling, northwest, data set, etc.) and